# On Giant's Shoulders:
# Effortless Weak to Strong by Dynamic Logits Fusion

**Chenghao Fan**[1,2,*]   **Zhenyi Lu**[1,2,*]   **Wei Wei**[1,2,†]   **Jie Tian**[1,2],   **Xiaoye Qu**[1],
**Dangyang Chen**[3],   **Yu Cheng**[4]

[1] School of Computer Science & Technology, Huazhong University of Science and Technology
[2] Joint Laboratory of HUST and Pingan Property & Casualty Research (HPL)
[3] Ping An Property & Casualty Insurance Company of China, Ltd.
[4] The Chinese University of Hong Kong
{facicofan,luzhenyi529}@gmail.com,{weiw,xiaoye}@hust.edu.cn
chendangyang273@pingan.com.cn,chengyu@cse.cuhk.edu.hk

> **"If I have seen further than others, it is by standing on the shoulders of giants."**
> *— Isaac Newton in 1675*

## Abstract

Efficient fine-tuning of large language models for task-specific applications is imperative, yet the vast number of parameters in these models makes their training increasingly challenging. Despite numerous proposals for effective methods, a substantial memory overhead remains for gradient computations during updates. *Can we fine-tune a series of task-specific small models and transfer their knowledge directly to a much larger model without additional training?* In this paper, we explore weak-to-strong specialization using logit arithmetic, facilitating a direct answer to this question. Existing weak-to-strong methods often employ a static knowledge transfer ratio and a single small model for transferring complex knowledge, which leads to suboptimal performance. To surmount these limitations, we propose a dynamic logit fusion approach that works with a series of task-specific small models, each specialized in a different task. This method adaptively allocates weights among these models at each decoding step, learning the weights through Kullback-Leibler divergence constrained optimization problems. We conduct extensive experiments across various benchmarks in both single-task and multi-task settings, achieving leading results. By transferring expertise from the 7B model to the 13B model, our method closes the performance gap by 96.4% in single-task scenarios and by 86.3% in multi-task scenarios compared to full fine-tuning of the 13B model. Notably, we achieve surpassing performance on unseen tasks. Moreover, we further demonstrate that our method can effortlessly integrate in-context learning for single tasks and task arithmetic for multi-task scenarios.

## 1 Introduction

In recent years, Large Language Models (LLMs) have shown impressive performance in a wide range of tasks [5, 36, 50–52, 54, 61, 62], including code generation [11, 45], mathematical reasoning [2, 37], tool-use abilities [46, 53], *etc*. However, training such LLMs requires substantial computational resources, often involving thousands of GPUs and processing trillions of tokens [30, 42], making the adaptation of the base model for new knowledge inefficient. To address these challenges, parameter-efficient tuning methods [12, 16, 23] have emerged, aiming to achieve comparable performance to full fine-tuning while reducing GPU memory requirements. However, challenges persist in tuning

---

[*]   Equal contribution.
[†]   Corresponding authors.

and deploying large-scale models on common hardware, as they still involve computation-intensive processes like gradient calculation and back-propagation. Furthermore, these methods may not be feasible when training data are private.

This inspires us to ask: *Can we fine-tune only small models and then transfer their knowledge to a much larger model without requiring additional gradient updates?* If we could fuse the strong capabilities of a scaled LLM with the specialized knowledge acquired by a small model during fine-tuning, it would yield the practical benefit of approximating the results achieved by fine-tuning a large model, but without the associated computational costs. However, it is non-trivial due to *the differences in representation width and layer numbers* between the small and large models.

Recently, Mitchell et al. [41] and Liu et al. [33] attempt to address this challenge by transferring knowledge from a Small Language Model (SLM) to its larger counterpart through simple logit arithmetic operations during decoding. For instance, using models from the Llama-2 family, they can transfer the fine-tuned knowledge from a 7B-scale model to a 13B-scale model by performing log probability algebra: Llama-2-base 13B + (Llama-2-chat 7B - Llama-2-base 7B), where the first term represents the base log probabilities and the term in parentheses denotes the behavioral delta [41]. This behavioral delta can be weighted to adjust the balance between the pretrained knowledge and the transferred fine-tuned knowledge.

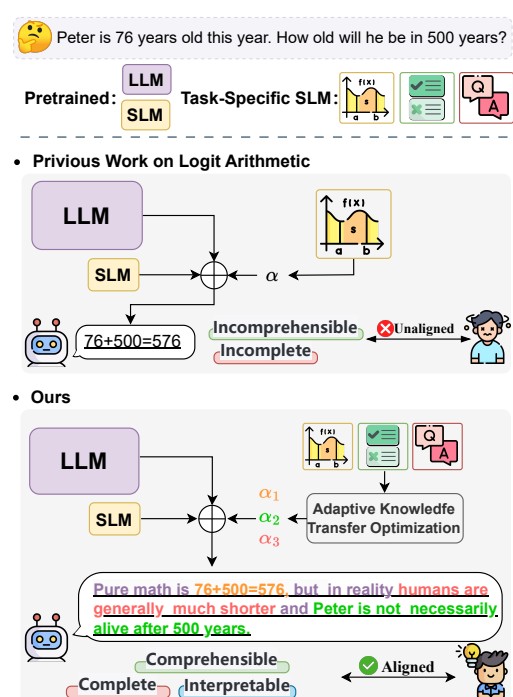

Figure 1: Comparison between our work and previous work. Previous methods only use pre-tuned parameters $\alpha$ to transfer knowledge from a single expert. In contrast, our method dynamically adjusts the proportion of knowledge transferred from multiple experts at each decoding step during inference.

Despite showing promise, logit arithmetic still exhibits a noticeable performance gap compared to directly fine-tuning large models, primarily due to two reasons: *Firstly*, they statically prespecifies the weight of behavioral delta at each decoding step identically. However, the importance of fine-tuned knowledge varies significantly across different tasks, inputs, and even different decoding steps. For instance, in a domain-specific question-answering process, we need more knowledge from fine-tuned small language models. Conversely, when decoding factual topics, we may need more knowledge from pretrained general models. *Secondly*, for unseen tasks, the lack of pre-adjusted weights for the behavioral delta prevents logit arithmetic from executing effectively, making it challenging to transfer tuned knowledge, especially for complex tasks. As shown in Figure 1, when answering "Peter is 76 years old this year. How old will he be in 500 years?", using only the math expert does not ensure the result aligns with factual accuracy. Additionally, these techniques often assume that experts are trained and tested on the same data distribution, ignoring the heterogeneity of data that may be encountered at test time, rendering any single expert insufficient.

In addressing these challenges, we reconsider the practice of logit arithmetic within a new framework. Specifically, we work with a set of finely tuned SLM experts, each specializing in different tasks. At each decoding step, we dynamically allocate weights among these task-specific SLMs. However, it is non-trivial to effectively determine suitable weights for diverse tasks and SLM experts, as tranditional approaches such as grid search [29] or combinatorial search [34] suffer from costly search processes. To practically automate weight allocation, we reframe the problem of weight search as a constrained distribution optimization problem. We tackle the fusion of multi-task knowledge by treating it as a centroid problem in Euclidean space using Kullback-Leibler divergence [24], which offers interpretability and explicit guidance.

We conduct thorough experiments on the LLaMA series to evaluate the effectiveness of our approach, applying our adaptive logit arithmetic method to task-specific fine-tuning in math, question-answering, summarization, and multi-domain tasks. We also analyze the performance of multi-model fusion across seen and unseen tasks. Additionally, we discussed the feasibility of combining our method with in-context learning for single tasks and with task arithmetic for multi-task scenarios.

In our work, we make several key contributions. Firstly, we reassess existing logit arithmetic methods, highlighting the significant impact of fusion weights and the limitations imposed by a single small model on test performance. Secondly, we introduce a novel approach that autonomously learns fusion weights through constraint optimization, approximating the compute-intensive results of fine-tuning a large base model. Lastly, we conduct comprehensive experiments to validate our method, demonstrating substantial improvements in performance, generalization capabilities, and robustness.

## 2 Related Work

### 2.1 Efficient Specialization

Specializing a pretrained model by fine-tuning for downstream tasks has become a primary paradigm [7, 9, 10, 44]. However, with the increasing scale of models, full fine-tuning has become impractical due to the large number of parameters, requiring costly training resources [5, 54].

To tackle this challenge, researchers have developed parameter-efficient tuning methods [28]. These methods aim to achieve performance similar to full fine-tuning while reducing the GPU memory required. They typically involve freezing the original weights and adding task-specific trainable modules [15, 22, 26], low-rank matrices [12, 16, 23], or bias [3]. Despite their benefits, these approaches still demand significant memory for updating gradients and might not be suitable for scenarios where training data are private or inaccessible. Another direction is model merging [17, 35, 38, 56, 58], which trains task-specific models on different domains, and then combines them into a single model at deployment with weight average [57], neuron permutation [1], interpolation [20, 39] or task vectors [17, 35, 59]. However, they typically suffer from parameter interference between different tasks and static optimal solutions may struggle with multiple task domains.

There is a growing research interest in specialized large models by eliciting existing knowledge, such as utilizing curated prompts via in-context learning [31] or employing small models to offer weak-to-strong supervision [6, 33, 41], generating weak labels or alignment signals. These methods are highly practical, requiring only the generated output (or logits). Our approach shares a similar weak-to-strong intuition and proposes adaptive strategies to achieve a better balance between leveraging the pretrained general knowledge and acquiring task-specific knowledge. Furthermore, our method can be extended to multi-task scenarios.

### 2.2 Weak-to-Strong Generation

Different from knowledge distillation [4, 14, 49], where a more capable teacher guides a student model, we explore using weaker models to teach stronger ones. Burns et al. [6] empirically demonstrate that weak supervisors can reliably elicit knowledge from much stronger models (*e.g.*, supervising GPT-4 with a GPT-2). Such approaches represent a promising direction, as it is more practical and affordable to specialize in a much smaller model. Contrastive Decoding [27] enhances outputs by leveraging differences between large and small LMs, as patterns like repetition and incoherence are often more pronounced in smaller ones. Speculative sampling [25, 48] speeds up inference using a lower-parameter version of the LLM as a draft model, exploiting the fact that draft models can approximate easier subtasks well. SpecInfer [40] goes further by employing a set of small models to generate drafts in parallel. Jin et al. [19] train a projector to map the parameter of the weak expert to a larger version, but the projector needs to be trained and suffers from poor generation. Ji et al. [18] focuses on utilizing small models to rephrase unaligned answers from LLMs into more aligned ones. Mitchell et al. [41] and Liu et al. [33] leverage logits from small, fine-tuned models to inject specific knowledge into the pretrained LLM with the same vocabulary. Our approach differs from methods that require pretrained small model adapters [48] or pre-adjusted parameters [33, 41]. Instead, we dynamically transfer the capabilities of small models to the large model at each decoding step, without needing access to the large model's parameters or training data.

# 3 Methodology

## 3.1 Problem Background

**Autoregressive Language Models** Modern autoregressive transformers generate continuations for input prompts token by token. Given a prompt $x_{1:k-1}$ (denoted as $x_{<k}$), the model computes the logits for the $k$-th token, represented as $M(x_k \mid x_{<k}) \in \mathbb{R}^{|V|}$, where V denotes the size of the vocabulary. A probability distribution $P(x_k \mid x_{<k})$ is then obtained through softmax normalization: $P(x_k \mid x_{<k}) = \text{softmax}(M(x_k \mid x_{<k}))$. The next token $x_k$ is subsequently sampled from this distribution, *i.e.*, $x_k \sim P(x_k \mid x_{<k})$.

**Distance Between Language Model Outputs Distribution** We can utilize the Kullback-Leibler(KL) divergence to measure the similarity of two distributions $P$ and $Q$ generated from two language models (with the same vocabulary), which can be viewed as the distance of the two language models:

$$D_{\text{KL}}(P||Q \mid x_{<k}) = \sum_{x \in V} P(x \mid x_{<k}) \log \frac{P(x \mid x_{<k})}{Q(x \mid x_{<k})} \tag{1}$$

If this is implied by the context, we will omit the conditioning on $|x_{<k}$ and simply use $D_{\text{KL}}(P||Q)$.

**Logit Arithmetic** Suppose we have two pretrained auto-regressive models with homogeneous architecture and the same vocabulary: a small model with parameter set $\theta^S$ and a large model with parameter set $\theta^L$. We aim to fine-tune the small model to obtain $\theta_{ft}^S$ and transfer this fine-tuning knowledge to the large models. Previous work [33, 41] transferred fine-tuned knowledge to a large model by designing arithmetic between logits, resulting in the output distribution $\tilde{P}$ for the large model as follows:

$$\tilde{P}(x_k|x_{<k}) = \text{softmax}(M^L(x_k|x_{<k}) + \alpha \cdot (M_{ft}^S(x_k|x_{<k}) - M^S(x_k|x_{<k}))) \tag{2}$$

where $M^L$, $M^S$, and $M_{ft}^S$ represent the logits of the large model, small model, and fine-tuned small model, respectively. Their corresponding normalized distributions are denoted by $P$, $Q$, and $Q_{ft}$. The detailed theoretical proof supporting this logit arithmetic formula is provided in Appendix B. Here, $\alpha$ is a pre-adjusted hyperparameter that controls the extent of knowledge transfer from the small model. Our analysis from Appendix C demonstrates that logit arithmetic attempts to approximate the shift $(\frac{Q_{ft}(x_k|x_{<k})}{Q(x_k|x_{<k})})^\alpha$ between the fine-tuned distribution and the pretrained distribution by controlling the parameter $\alpha$ before inference.

## 3.2 Adaptive Knowledge Transfer Optimization

However, using a pre-defined $\alpha$ presents the issue that the resulting LLM's distribution $\tilde{P}$ has a fixed trajectory, which may not be suitable for every step in decoding. So we need a dynamic $\alpha$ in the decoding steps. Nonetheless, solving this problem is non-trivial, so we attempt to transform it into an equivalent, optimizable objective. Here, we use the distance between language model outputs as defined in Equation (1) to represent the shift between distributions. We assume that, for different model sizes, at each decoding step, the distance between the fine-tuned model outputs and pretrained model output is the same:

$$D_{\text{KL}}(\tilde{P}||P \mid x_{<k}) \approx D_{\text{KL}}(Q_{ft}||Q \mid x_{<k}), \quad D_{\text{KL}}(P||\tilde{P} \mid x_{<k}) \approx D_{\text{KL}}(Q||Q_{ft} \mid x_{<k}) \tag{3}$$

Following the assumption in Equation (3), we optimize the following expression at each decoding step:

$$\arg \min_{\tilde{P}} \left[ \left( D_{\text{KL}}(\tilde{P}||P) - D_{\text{KL}}(Q_{ft}||Q) \right)^2 + \left( D_{\text{KL}}(P||\tilde{P}) - D_{\text{KL}}(Q||Q_{ft}) \right)^2 \right] \tag{4}$$

Based on this optimization problem, we can adaptive transfer the knowledge of the corresponding SLM expert to the large model at each decoding step in a single-task setting. Our method has been proven to be effective in experiments. To guarantee symmetry, a symmetrical term is incorporated:

### 3.3 Extending to the Fusion of Multiple SLMs

When dealing with complex tasks or new domains, a general LLM may lack the necessary expertise, and a single SLM might not provide sufficient specialized knowledge due to the capacity gap. To migrate this challenge, our method can be extended to fuse multiple SLMs and leverage their knowledge to compensate for the shortcomings in individual domain-specific knowledge. We fine-tune the SLM $\theta^S$ on each domain to obtain multiple task-specific experts $\{\theta_t^S\}_{t=1}^T$, making it easier to acquire knowledge and dynamic fused to the LLM. During decoding, knowledge from these domain-specific SLMs is transferred simultaneously to the LLM. We modify the Equation (3) as follows:

$$D_{\mathrm{KL}}(\tilde{P} \parallel P) \approx D_{\mathrm{KL}}\left(\mathrm{Joint}(\{Q_{1...T}\}) \parallel Q\right), \quad D_{\mathrm{KL}}(P \parallel \tilde{P}) \approx D_{\mathrm{KL}}\left(Q \parallel \mathrm{Joint}(\{Q_{1...T}\})\right) \quad (5)$$

where $Q_t$ represents the distribution of $\theta_t^S$ from the $t$-th domain, $\mathrm{Joint}(\{Q_{1...T}\})$ represents the distribution of the combined knowledge of $Q_1, Q_2, ..., Q_T$. When we impose constraints like Equation (5), it attempts to align the joint distributions of the logits from the domain-specific small models. However, the distributions of the logits from the domain-specific small models are usually not independent of each other, so their joint distribution cannot be estimated through simple multiplication.

Due to the difficulty in obtaining a joint distribution for multiple expert models, we decompose the joint distribution constraint problem into a multi-object marginal distribution optimization. The transformation process we prove in detail in Appendix E. By aligning the distributions of each domain-specific small model, we can infer an approximate optimal solution within the extension of Equation (2), as shown by:

$$\arg\min_{\tilde{P}} \sum_{t=1}^T \left[ \left( D_{KL}(\tilde{P}\|P) - D_{KL}(Q_t\|Q) \right)^2 + \left( D_{KL}(P\|\tilde{P}) - D_{KL}(Q\|Q_t) \right)^2 \right]$$

$$\text{where} \quad \tilde{P} = \mathrm{softmax}\left[ M^L(x_k|x_{<k}) + \sum_{t=1}^T \alpha_t \left( M_t^S(x_k|x_{<k}) - M^S(x_k|x_{<k}) \right) \right] \quad (6)$$

Here we use $M_t^S$ to represent the logit of $t$-th expert. Our algorithm is outlined in pseudo-code in Algorithm 1 in Appendix F.

Intuitively, this projects the KL divergences between the logits into Euclidean space and finds a central point with the minimum distance sum as the best KL value. This optimal KL value corresponds to the output distribution of the large model with multi-domain knowledge. In our experiments, we optimize $\alpha \in \mathbb{R}^T$ to obtain the optimal KL value. Generally, $\alpha$ is between 0 and 2. In a multi-task setting, we can accelerate the optimization process by optimizing the boundaries for only one expert, restricting only one SLM expert to be applied at the current decoding step. We will provide a more detailed explanation of this in our experiments.

## 4 Experiments

In this paper, we use the LLaMA2 [55] family of models to test our method, which contains the same vocabulary, enabling us to employ logits arithmetic easily. Here, we use TinyLLaMA-1.1B [60] and LLaMA2-7B as our pretrained small models, and LLaMA2-13B as the large model for transfer. We conduct tests in single-domain and multi-domain settings to demonstrate that our method can effectively transfer knowledge to the large model in both scenarios.

### 4.1 Datasets

Following the setting in Liu et al. [33], we evaluate on the following datasets: mathmetical reasoning (GSM8K [8]); factual accuracy (TruthfulQA [32]); realistic knowledge (TriviaQA [21]); multi-domain general knowledge (MMLU benchmark [13]); summarization (CNN-DailyMail (CNN/DM) [47]). All datasets are tested using a 0-shot setting. Detailed information is provided in Appendix G.

### 4.2 Implementation Details

For all tasks except TruthfulQA, we construct prompts based on the task type and employ supervised instruction tuning to train each task expert. Following the previous work [33], we use "Llama-2-7b-chat-hf" as the 7B expert for TurhfulQA, and TinyLLaMA-chat-version as the 1.1B expert for

Table 1: Performance on single-task scenarios. **Bold** numbers indicate the best-performing model transferred from the same size. Underlines indicate whether the method outperforms the expert model being used. Notably, we are unable to obtain the LoRA adapter for LLAMA2-chat version. Therefore, we set the LoRA Tuning for the 13B model on TruthfulQA to match the same values as Full Fine-Tuning, *e.g.*, 61.93.

| Model | GSM8K (EM.) | TruthfulQA (Acc.) | TriviaQA (EM.) | CNN/DM (Rouge 2.) | MMLU (Acc.) | Avg. |
|---|---|---|---|---|---|---|
| *13B* | | | | | | |
| **Base Model** | 6.90 | 46.13 | 36.44 | 8.94 | 51.25 | 29.93 |
| **Full Fine-tuning** | 47.23 | 61.93 | 56.36 | 15.50 | 57.94 | 47.79 |
| **LoRA Tuning** | 41.54 | 61.93 | 61.89 | 17.18 | 60.46 | 48.60 |
| Transfer from *1.1B* | | | | | | |
| **Full Fine-tuning** | 12.51 | 29.01 | 33.66 | 14.22 | 37.26 | 25.33 |
| **Proxy Tuning** | 16.91 | 31.48 | 48.74 | 13.23 | 39.88 | 31.74 |
| **Ours** | **18.27** | 37.05 | **53.81** | **14.48** | 48.32 | **34.86** |
| Transfer from *7B* | | | | | | |
| **Full Fine-tuning** | 37.07 | 60.02 | 52.10 | 15.21 | 56.23 | 44.13 |
| **Proxy Tuning** | 37.68 | 61.02 | 52.81 | 14.37 | 56.24 | 44.43 |
| **Ours** | **39.34** | **61.56** | **57.11** | **15.31** | **57.15** | **46.09** |

Table 2: Performance on multi-task scenarios. "Base" denotes an untrained model. "Multi-Task Tuning" refers to models trained using data mixing. **Bold** numbers represent the best-performing multi-task models among those using experts of the same size. The leftmost "Avg." represents the average performance of Seen Tasks and Unseen Tasks (57 tasks in MMLU), calculated by averaging the mean performance on Seen Tasks and the performance on MMLU.

| Model | Avg. | Seen Task | | | | | Unseen Task |
|---|---|---|---|---|---|---|---|
| | | GSM8K | TruthfulQA | TriviaQA | CNN/DM | Avg. | MMLU |
| *Pre-trained model* | | | | | | | |
| **1.1B (Base)** | 18.29 | 2.04 | 27.41 | 9.6 | 7.21 | 11.56 | 25.02 |
| **7B (Base)** | 29.16 | 3.80 | 30.96 | 36.7 | 8.81 | 20.06 | 38.26 |
| **13B (Base)** | 37.93 | 6.89 | 46.13 | 36.5 | 8.94 | 24.61 | 51.25 |
| *Multi task with tuning* | | | | | | | |
| **13B (Multi-Task Tuning)** | 45.68 | 39.03 | 44.39 | 62.79 | 16.95 | 40.78 | 50.58 |
| *Single task with tuning* | | | | | | | |
| **1.1B-Expert (GSM8K)** | 18.50 | 12.51 | 25.38 | 6.12 | 7.75 | 12.69 | 24.30 |
| **1.1B-Expert (TruthfulQA)** | 16.15 | 2.81 | 29.01 | 2.83 | 7.23 | 10.47 | 21.82 |
| **1.1B-Expert (TriviaQA)** | 21.72 | 3.26 | 26.25 | 33.66 | 8.03 | 17.80 | 25.63 |
| **1.1B-Expert (CNN/DM)** | 12.98 | 2.73 | 26.39 | 1.65 | 14.22 | 11.24 | 14.73 |
| *Multi-task transfer from 1.1B* | | | | | | | |
| **1.1B (Multi-Task Tuning)** | 23.90 | 14.40 | 25.76 | **35.05** | 14.26 | **22.37** | 25.42 |
| **Ours** | **32.59** | **18.65** | **36.33** | 18.84 | 9.38 | 20.80 | **44.38** |
| *Single task with tuning* | | | | | | | |
| **7B-Expert (GSM8K)** | 34.14 | 37.07 | 36.04 | 20.41 | 11.19 | 26.18 | 42.10 |
| **7B-Expert (TruthfulQA)** | 33.23 | 8.26 | 60.02 | 0.17 | 11.02 | 19.86 | 46.61 |
| **7B-Expert (TriviaQA)** | 29.35 | 4.62 | 33.66 | 52.10 | 10.30 | 25.17 | 33.52 |
| **7B-Expert (CNN/DM)** | 22.29 | 4.39 | 34.57 | 0.19 | 15.21 | 13.59 | 30.98 |
| *Multi-task transfer from 7B* | | | | | | | |
| **7B (Multi-Task Tuning)** | 34.89 | 34.72 | 33.28 | **51.54** | **16.30** | **33.96** | 35.82 |
| **Ours** | **39.42** | **34.87** | **42.25** | 22.48 | 10.52 | 27.53 | **51.31** |

TurhfulQA. For full fine-tuning, we set the batch size to 128, learning rate to 2e-5, optimizer to Adam. For LoRA tuning, we set the rank to 64, learning rate to 1e-4, optimizer to Adam. We train for 3 epochs. For multi-task tuning, we perform full fine-tuning using a mixed-seed training dataset.

During inference, we use greed decoding and set batch size to 256, top_p to 1.0 and temperature to 0.05. To accelerate inference, we use VLLM[3] , synchronizing the signals of logits during logit arithmetic to achieve efficient inference. All experiments are performed on H100 GPUs.

---

[3]https://github.com/vllm-project/vllm

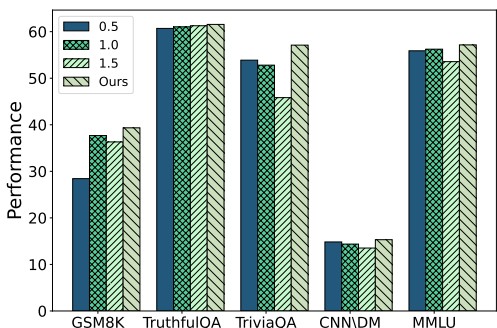

Figure 2: Compare the pre-defined $\alpha$ with the dynamic $\alpha$ for different tasks.

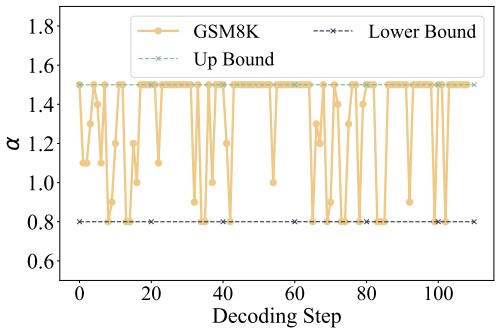

Figure 3: The variation of $\alpha$ in knowledge transfer for the GSM8K expert. Lower Bound and Upper Bound represent the minimum and maximum values obtained during the optimization process. In the

## 4.3 Baselines

To demonstrate the effectiveness of our method in efficiently transferring knowledge from a small model to a large model, we compare it against a small model fine-tuned in a single domain and a large model without fine-tuning. We also report the performance of a large model fine-tuned on a single domain or fine-tuned using LoRA to demonstrate the feasibility of our method as a replacement for fine-tuning approaches. Additionally, we compare our method with Proxy Tuning (setting $\alpha = 1.0$ as in the original work) to highlight the superiority of our method in the transfer process. We use the LLaMA2-chat model as the expert for TruthfulQA (therefore, there is no corresponding LoRA model). For other experts, we use models fine-tuned on the respective training sets. In multi-task scenarios, we follow the common training practice of mixing the training data, with the results serving as our multi-task baseline. More details can be found in the Appendix G.

## 4.4 Performance on Single-Task Scenarios

As shown in Table 1, our method improves upon the original experts when transferring knowledge from the 1.1B and 7B models to the 13B model. Across all tasks, our method transfers knowledge from 1.1B and 7B experts to the 13B model, achieving performance improvements of 37.6% and 4.4% over the original experts, respectively. This demonstrates that our method can leverage the existing knowledge in larger models to enhance the knowledge learned by the expert small models. Notably, we observe significant improvements on GSM8K and TriviaQA compared to the experts alone, indicating that our method effectively transfers expert knowledge while leveraging the inherent capabilities of the large model, such as stronger reasoning abilities and a richer knowledge base.

Our method, which transfers knowledge from 1.1B and 7B experts, can close the performance gap by 72.9% and 96.4%, respectively, compared to 13B Full Fine-Tuning, achieving improvements of 6.5% and 3.8% over Proxy Tuning. Compared to 13B Full LoRA Tuning, our method can close the gap by 71.7% and 90.7%, respectively, when transferring knowledge from 1.1B and 7B experts. Additionally, it is worth noting that our method outperforms the 13B Full Fine-Tuning results on TriviaQA. Since our approach only requires fine-tuning a smaller model, it demands less computational and memory resources compared to fine-tuning a large model, making our method highly promising.

## 4.5 Performance on Multi-Task Scenarios

In the multi-task scenario, we categorize the multi-domain MMLU task as an unseen task and the other four tasks as seen tasks. We then calculate the average performance on seen and unseen tasks to evaluate the overall generalization capability of our model. As shown in Table 2, our method achieves a 36.35% and 13.37% improvement over directly fine-tuning on 1.1B and 7B, respectively, using multi-domain fine-tuning. Furthermore, our results show that transferring knowledge from 7B outperforms 13B overall by 3.93%. Specifically, the performance improvement is 11.9% for seen tasks and 0.1% for unseen tasks. This indicates that our approach can alleviate conflicts between multiple

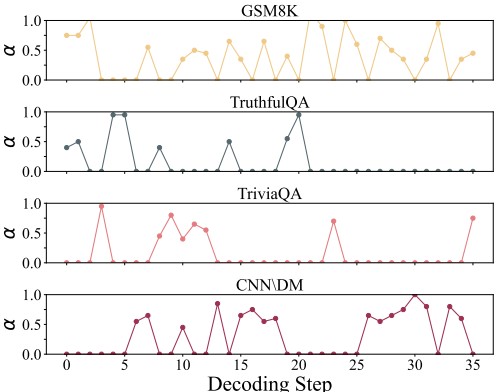

Figure 4: The variation of $\alpha$ for the four experts during knowledge transfer on an unseen task (MMLU: abstract algebra).

Table 3: The time required to train or inference 1,000 data points on a single GPU. In the inference section, values in parentheses show the factor by which the inference speed is slower compared to the 13B FFT model. "FFT" denotes Full Fine-Tuning, and "LT" denotes LoRA Tuning. Our 1.1B/7B expert model use full fine-tuning.

| Model | Training | Inference |
|---|---|---|
| 13B FFT | 1176s | **60s** |
| 13B LT | 836s | **60s** |
| Proxy Tuning (from 1.1B) | **128s** | 142s (×2.36) |
| Ours (from 1.1B) | **128s** | 150s (×2.5) |
| Proxy Tuning (from 7B) | 588s | 158s (×2.63) |
| Ours (from 7B) | 588s | 166s (×2.76) |

tasks to a certain extent and improve out-of-domain generalization by leveraging the capabilities of the large model itself.

In multi-task settings, our method achieves a performance improvement of 71.3% and 86.3% over 13B multi-task tuning when transferring knowledge from 1.1B and 7B, respectively. It can be observed that our method performs worse in domain-specific tasks compared to multi-task tuning, *e.g.*, TriviaQA. This is primarily because we cannot access the data needed by the experts, resulting in a bottleneck in handling task conflicts. In the future, we will explore more effective ways to effortlessly transfer multiple small expert models to a large model. Notably, Proxy Tuning is designed for single-domain scenarios with a known test distribution. It struggles at test time with inputs from unknown distributions, making it difficult to use in multi-task scenarios. However, our method achieves the best results on unseen tasks, demonstrating its effectiveness in enhancing the generalization ability of large models on new tasks. Moreover, our method can effortlessly transfer knowledge from pretrained experts across multiple domains to the large model without requiring access to their training data.

## 5 Analysis

### 5.1 How $\alpha$ control the knowledge transfer from the small models?

We aim to understand how $\alpha$ dynamically changes in single-task and multi-task scenarios, and what patterns it exhibits across different tasks. As shown in Figure 2, we compare our method with predefined $\alpha$ values of 0.5, 1.0, and 1.5. It can be observed that our method consistently outperforms the predefined $\alpha$ settings, demonstrating the effectiveness of adaptive adjustment at each decoding step. Additionally, the predefined method exhibits significant variation across different tasks, whereas our method addresses this issue and avoids the extensive effort required for parameter tuning.

As shown in Figures 3 and 4, we illustrate the variation of $\alpha$ for a single expert and multiple experts, respectively. In Figure 3, it can be observed that $\alpha$ fluctuates between the lower bound and upper bound. At the lower bound, less expert knowledge is transferred, and it is more important to retain the large model's inherent capabilities. Conversely, at the upper bound, more expert knowledge is transferred, requiring more GSM8K-related capabilities for those steps. In Figure 4, when solving the "abstract algebra" problem in MMLU using four experts, it is evident that GSM8K transfers more knowledge overall, especially in the initial steps. Since "abstract algebra" is a mathematically oriented problem, these results demonstrate that our method can effectively select and transfer expert knowledge during the decoding steps.

### 5.2 Efficiency Analysis

In a batch inference with a size of $B$, Proxy Tuning requires a complexity of approximately $O(BV)$ for each decoding step. In contrast, our method requires $O(nBV)$ complexity, where $n$ ($\leq 20$) is the number of parameter searches. Since $n \ll V$, our complexity is comparable to Proxy Tuning.

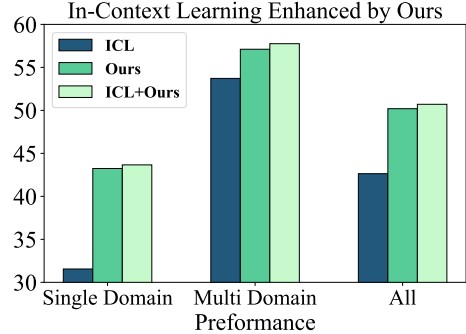

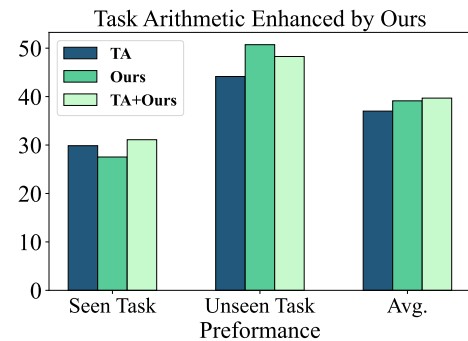

(a) Enhance In-Context Learning (ICL) with our method from transferring 7B expert knowledge.

(b) Enhance Task Arithmetic (TA) with our method from transferring 7B expert knowledge.

Figure 5: Enhance in-context learning and task arithmetic using our method.

In multi-task settings, performing parameter searches for each expert and then merging them would result in exponential growth, with a complexity of $O(n^T BV)$. To optimize this process, we avoid merging parameters between experts at each step and instead select a single expert for knowledge transfer. This reduces the algorithm complexity to $O(nTBV)$. Additionally, we can constrain the parameter search range for each expert to achieve better efficiency and performance.

Here, we complete the training and inference on an H100 GPU for 1000 data, while recording the time taken. As shown in Table 3, our method achieves similar efficiency to Proxy Tuning during inference. Due to the need for inference and communication between multiple models, our approach is approximately 2.5 times slower than direct inference with a 13B model. However, our method only requires training a small model, which provides a significant advantage in training efficiency.

### 5.3 Comparison the Weak-to-strong Approach with Other Methods

**Comparison against In-Context Learning**     Enhancing in-context learning with logits arithmetic can improve large language models. Since both methods work at the token level and need only black-box access, they can be combined to boost model performance. We categorize tasks as either Single-Domain (excluding MMLU) or Multi-Domain (MMLU). The "All" category averages the results of both. Notably, we use 5-shot information as in-context examples, and we apply different few-shot samples and prompts for various MMLU tasks. In our experiments, we enhance a 13B model's performance using a 7B expert and compare it to the 13B model using in-context learning. We present the results in Figure 5a. We observed that our method outperforms 5-shot in-context learning on both Single-Domain and Multi-Domain tasks. Furthermore, when combined with in-context learning, our method shows a significant improvement on Multi-Domain tasks, resulting in an overall (All) increase of 18.3%. Specifically, combining our method with in-context learning does not significantly impact performance on Single-Domain tasks, indicating that our method alone has already achieved capabilities similar to in-context learning for individual tasks.

**Comparison against Task Arithmetic**     Task Arithmetic and logits arithmetic are similar in principle, both adjusting the shift between the expert and base model to control knowledge transfer. However, logits arithmetic isn't constrained by parameter size and can merge across model levels without needing access to specific parameters. Specifically, our method can be applied to combine experts from task arithmetic, integrating both approaches for multi-task learning. We use the same setup as our multi-task scenarios, treating "All" as the average of Seen and Unseen Tasks. In figure 5b, our method performs well on Unseen Tasks, resulting in an overall improvement of 5.7% compared to task arithmetic. When we combine our method with task arithmetic, we see improvements over task arithmetic alone, achieving the highest overall performance, with increases of 7.2% and 1.5% over task arithmetic and our method alone, respectively.

# 6 Conclusions

In this paper, we introduce a dynamic logit fusion approach in weak-to-strong specialization, which utilizes a series of task-specific small models and allows for adaptive weight allocation among them. Through extensive experiments, we have demonstrated the effectiveness of our approach in both single-task and multi-task settings across various benchmarks. By transferring expertise from the 7B model to the 13B model, we have achieved significant performance improvements, closing the performance gap by 96.4% in single-task scenarios and by 86.3% in multi-task scenarios compared to full fine-tuning of the larger model. Our method also shows promising results on unseen tasks and can integrate in-context learning for single tasks and task arithmetic for multi-task scenarios.

# 7 Acknowledgments

This work was supported in part by the National Natural Science Foundation of China under Grant No. 62276110, No. 62172039 and in part by the fund of Joint Laboratory of HUST and Pingan Property & Casualty Research (HPL). We thank the Shanghai AI Laboratory for supporting GPU resources. The authors would also like to thank the anonymous reviewers for their comments on improving the quality of this paper.

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

# A    Model Architecture Diagram

As illustrated in Figure 6, our approach dynamically calculates the fusion weights of the logits of SLMs at each decoding step, transferring the knowledge from multiple experts to LLM.

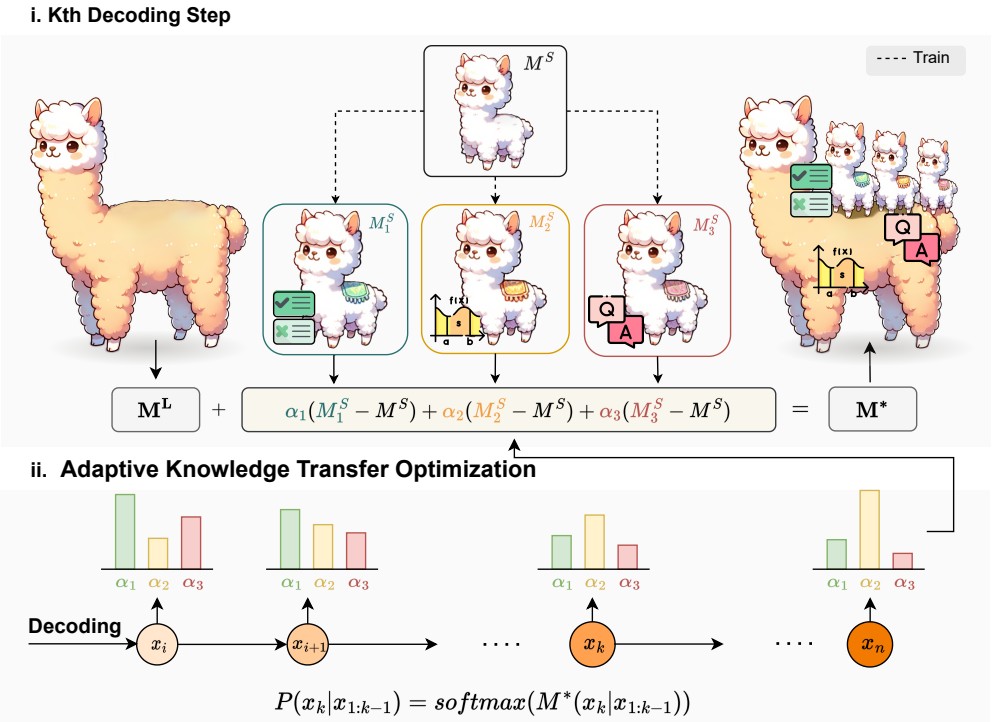

Figure 6: The architecture of our method. The small llama represents the small model, while the large llama represents the large model. $M^S/M^L$ denotes the logits of the small/large language model. $M_t^S$ represents the logits of a small expert language model for task $t$. The lower part of the figure illustrates our optimization in the decoding process, where each circle represents a decoding step. The upper part of the figure shows how our method transfers the knowledge of experts in the $k$th step. At each decoding step, our method dynamically adjusts the $\{\alpha_t\}_{t=1}^T$ value for each expert, transferring knowledge from the small models to the larger model.

# B    Proof of Logic Arithmetic

Reinforcement Learning from Human Feedback (RLHF) is commonly employed in optimizing language models, where the learning process involves maximizing a reward objective while penalizing the KL divergence between the learned policy and a reference policy with a coefficient $\beta$:

$$\max_{\pi_\theta} \mathbb{E}_{x\sim\mathcal{D},y\sim\pi_\theta(y|x)}[r_\phi(x,y)] - \beta D_{\mathrm{KL}}\left[\pi_\theta(y\mid x)\|\pi_{\mathrm{ref}}(y\mid x)\right] \tag{7}$$

Following the DPO [43] framework, we can reformulate the above objective as:

$$\max_{\pi_\theta} \mathbb{E}_{x\sim\mathcal{D}}\mathbb{E}_{y\sim\pi_\theta(y|x)}\left[r_\phi(x,y)\right] - \mathbb{E}_{x\sim\mathcal{D}}\mathbb{E}_{y\sim\pi_\theta(y|x)}\left[\beta\log\frac{\pi(y\mid x)}{\pi_{\text{ref}}(y\mid x)}\right]$$

$$= \max_{\pi_\theta} \mathbb{E}_{x\sim\mathcal{D}}\mathbb{E}_{y\sim\pi_\theta(y|x)}\left[r_\phi(x,y) - \beta\log\frac{\pi(y\mid x)}{\pi_{\text{ref}}(y\mid x)}\right]$$

$$= \min_{\pi} \mathbb{E}_{x\sim\mathcal{D}}\mathbb{E}_{y\sim\pi(y|x)}\left[\log\frac{\pi(y\mid x)}{\pi_{\text{ref}}(y\mid x)} - \frac{1}{\beta}r_\phi(x,y)\right]$$

$$= \min_{\pi} \mathbb{E}_{x\sim\mathcal{D}}\mathbb{E}_{y\sim\pi(y|x)}\left[\log\frac{\pi(y\mid x)}{\pi_{\text{ref}}(y\mid x)} - \log\exp\left(\frac{1}{\beta}r_\phi(x,y)\right)\right]$$

$$= \min_{\pi} \mathbb{E}_{x\sim\mathcal{D}}\mathbb{E}_{y\sim\pi(y|x)}\left[\log\frac{\pi(y\mid x)}{\pi_{\text{ref}}(y\mid x)\exp\left(\frac{1}{\beta}r_\phi(x,y)\right)}\right] \tag{8}$$

$$= \min_{\pi} \mathbb{E}_{x\sim\mathcal{D}}\mathbb{E}_{y\sim\pi(y|x)}\left[\log\frac{\pi(y\mid x)}{\pi_{\text{ref}}(y\mid x)\exp\left(\frac{1}{\beta}r_\phi(x,y)\right)\frac{1}{Z(x)}Z(x)}\right]$$

$$= \min_{\pi} \mathbb{E}_{x\sim\mathcal{D}}\mathbb{E}_{y\sim\pi(y|x)}\left[\log\frac{\pi(y\mid x)}{\frac{1}{Z(x)}\pi_{\text{ref}}(y\mid x)\exp\left(\frac{1}{\beta}r_\phi(x,y)\right)} - \log Z(x)\right]$$

where $Z(x) = \sum_y \pi_{\text{ref}}(y\mid x)\exp\left(\frac{1}{\beta}r_\phi(x,y)\right)$ is the partition function. Note that the partition function is a function of only $x$ and the reference policy $\pi_{\text{ref}}$, but does not depend on the policy $\pi_\theta$. We can now define

$$\pi^*(y\mid x) = \frac{1}{Z(x)}\pi_{\text{ref}}(y\mid x)\exp\left(\frac{1}{\beta}r_\phi(x,y)\right), \tag{9}$$

Then the equation is convert to

$$\min_{\pi}\mathbb{E}_{x\sim\mathcal{D}}\left[\mathbb{E}_{y\sim\pi(y|x)}\left[\log\frac{\pi(y|x)}{\pi^*(y|x)}\right] - \log Z(x)\right] = \\ \min_{\pi}\mathbb{E}_{x\sim\mathcal{D}}\left[D_{\text{KL}}(\pi(y|x)\,||\,\pi^*(y|x)) - \log Z(x)\right] \tag{10}$$

Hence we have the optimal solution:

$$\pi(y|x) = \pi^*(y|x) = \frac{1}{Z(x)}\pi_{\text{ref}}(y|x)\exp\left(\frac{1}{\beta}r_\phi(x,y)\right) \tag{11}$$

for all $x \in \mathcal{D}$.

According to Rafailov et al. [43], reward models can be reparameterized as: $r(x,y) = \beta\log\frac{\pi(y|x)}{\pi_{\text{ref}}(y|x)}$ for some model $\pi(y\mid x)$ and a given reference model $\pi_{\text{ref}}(y\mid x)$. Additionally, given a pretrained language model $\pi$ and its fine-tuned counterpart $\pi_{\text{ft}}$, the following relationship holds,

$$\pi_{\text{ft}}(y\mid x) = \pi(y\mid x)\exp(\underbrace{\log\frac{\pi_{\text{ft}}(y\mid x)}{\pi(y\mid x)}}_{\text{Implicit reward}}) = \pi(y\mid x)\exp(\frac{1}{\beta}r_{\text{ft}}(x,y)) \tag{12}$$

This indicates that the fine-tuned model $\pi_{\text{ft}}$ can be interpreted as the solution to constrained reinforcement learning problem with a constraint on the pre-trained model. Consequently, the theoretical framework is applicable to any fine-tuned model, providing an RL-based interpretation of fine-tuning.

We can further introduce an $S$-size implicit reward for $L$-size finetuned models:

$$\pi_{\text{ft}}^L(y \mid x) = \frac{1}{Z(x)} \pi^L(y \mid x) \exp\left(\frac{1}{\beta} r_{\text{ft}}^S(x, y)\right)$$

$$\propto \pi^L(y \mid x) \exp\left(\log \frac{\pi_{\text{ft}}^S(y \mid x)}{\pi^S(y \mid x)}\right) \tag{13}$$

$$\propto \pi^L(y \mid x) \frac{\pi_{\text{ft}}^S(y \mid x)}{\pi^S(y \mid x)}$$

By taking the logarithms, we derive the following logit arithmetic formula:

$$M_{\text{ft}}^L(x) \propto M^L(x) + (M_{\text{ft}}^S(x) - M^S(x)) \tag{14}$$

where $M^L$, $M^S$, and $M_{ft}^S$ represent the logits of the large model, small model, and fine-tuned small model, respectively. This completes the derivation of the Equation (2).

## C   Proof of Method in Section 3.1

Based on the definitions of the probability distributions $Q$, $Q_{ft}$, $P$, and $\tilde{P}$, the model logits outputs $M^S$, $M_{ft}^S$, and $M^L$ satisfy the following relationships:

$$\begin{aligned}
Q(x_k \mid x_{<k}) &= \text{softmax}(M^S(x_k|x_{<k})) \\
Q_{ft}(x_k \mid x_{<k}) &= \text{softmax}(M_{ft}^S(x_k|x_{<k})) \\
P(x_k \mid x_{<k}) &= \text{softmax}(M^L(x_k|x_{<k})) \\
\tilde{P}(x_k|x_{<k}) &= \text{softmax}(M^L(x_k|x_{<k}) + \alpha \cdot (M_{ft}^S(x_k|x_{<k}) - M^S(x_k|x_{<k})))
\end{aligned} \tag{15}$$

Suppose there are vectors $x$ and $y$ of dimension $n$. The relationship between them is as follows:

$$y = softmax(x) = \left[\frac{e^{x_i}}{\sum_{j=1}^n e^{x_j}}\right]_{i=n}^n = \left[\frac{1}{\sum_{j=1}^n e^{x_j - x_i}}\right]_{i=n}^n \tag{16}$$

For each value $x_i$, as $x_i$ increases, $x_j - x_i$ decreases, and $\frac{1}{e^{x_j - x_i}}$ increases. Therefore, the corresponding $y_i$ is positively correlated with $x_i$.

So we can know that $\ln y \propto y \propto x$. Therefore, based on Equation (15), we can derive the following relationship:

$$\begin{aligned}
\ln Q(x_k \mid x_{<k}) &\propto M^S(x_k|x_{<k}) \\
\ln Q_{ft}(x_k \mid x_{<k}) &\propto M_{ft}^S(x_k|x_{<k}) \\
\ln P(x_k \mid x_{<k}) &\propto M^L(x_k|x_{<k}) \\
\ln \tilde{P}(x_k|x_{<k}) &\propto (M^L(x_k|x_{<k}) + \alpha \cdot (M_{ft}^S(x_k|x_{<k}) - M^S(x_k|x_{<k})) \\
&\propto \ln P(x_k \mid x_{<k}) + \alpha \cdot (\ln Q_{ft}(x_k \mid x_{<k}) - \ln Q(x_k \mid x_{<k})) \\
&\propto \ln\left(P(x_k \mid x_{<k}) \cdot \left(\frac{Q_{ft}(x_k \mid x_{<k})}{Q(x_k \mid x_{<k})}\right)^\alpha\right)
\end{aligned} \tag{17}$$

Thus, logit arithmetic can be viewed as a product-of-experts ensemble as follows:

$$\tilde{P}(x_k \mid x_{<k}) \propto P(x_k \mid x_{<k})\left(\frac{Q_{ft}(x_k \mid x_{<k})}{Q(x_k \mid x_{<k})}\right)^\alpha \tag{18}$$

Essentially, it assumes a predefined distribution shift $\left(\frac{Q_{ft}(x_k|x_{<k})}{Q(x_k|x_{<k})}\right)^\alpha$ between the fine-tuned and pretrained distributions of a small model, which can construct the trajectory of $\tilde{P}$. It then attempts to approximate the shift for the fine-tuned and pretrained distributions of the large model $\frac{\tilde{P}(x_k|x_{<k})}{P(x_k|x_{<k})}$.

## D    Efficiency Analysis of the Forward Process

In section 5.2, we conducted an Efficiency Analysis of logit Arithmetic. To better illustrate our efficiency, we further analyze the overall efficiency of our method here.

Overall, during a single forward pass, **our method has a similar time complexity to the static method**. Given: current sequence length $s$, large model dimension $h_L$, small model dimension $h_S$, number of layers in the large model $L_1$, number of layers in the small model $L_2$, batch size $B$, vocabulary size $V$, number of searches per decoding step $n$. Assume the FLOPs for a single forward pass of the large model and the small model are $FLOPs_L$ and $FLOPs_S$, respectively. The FLOPs can be calculated as: $FLOPs_L = L_1 * (12Bsh_L^2 + 2Bs^2h_L) + Bsh_LV$ ,and $FLOPs_S = L_2 * (12Bsh_S^2 + 2Bs^2h_S) + Bsh_SV$ (here we ignore the kv cache). Therefore, the FLOPs for a single forward pass of our method on a single task is: $FLOPs_L + 2 * FLOPs_S + nBV$. Among these, only the $nBV$ term ($n \leq 20$) corresponds to the additional computational cost of our method, which is much smaller compared to the previous term and can be considered negligible in the overall time. Additionally, in our efficiency analysis, as shown in Table 3, our method is only 0.008 seconds slower per sample compared to the static method, which is negligible.

## E    Proof for the Fusion of Multiple SLMs Scenario

This section mainly explains how we extend the transfer problem to multiple small models. When transferring the knowledge of multiple expert SLMs to a LLM, we consider the following two aspects: 1. The fusion of knowledge from different domain experts. 2. The transfer of knowledge from SLM to LLM, i.e., the transfer of knowledge from a single expert, which was discussed in Section 3.2. Intuitively, we first focus on the fusion of different domain experts' knowledge before performing the transfer. Here, we define the distribution of the combined knowledge of these small models as $J$. **Therefore, we aim to achieve** $D_{KL}(P||\tilde{P}) = D_{KL}(Q||J)$.

Since solving for $J$ is difficult, we propose constraining it based on the relationship between $J$ and $\{Q_i\}$ to approximate it. Here, we can transform $D_{KL}(Q||J)$ into $D_{KL}(Q||Q_i) + C_J(Q_i)$, where $C_J(Q_i)$ is the bias function from $Q_i$ to $J$. When we approximate $J$ as the centroid of $\{Q_i\}$ on the KL-constrained plane, we can implicitly solve these bias functions. According to the definition of the centroid, $J$ can be solved by minimizing the sum of the squared distances to each point, as shown below:

$$\arg\min_J \sum_{i=1}^{T} (D_{KL}(Q \parallel J) - D_{KL}(Q \parallel Q_i))^2 \tag{19}$$

Since our goal is $D_{KL}(P \parallel \tilde{P}) = D_{KL}(Q||J)$, substituting this into our equation gives us our final optimization objective:

$$\arg\min_{\tilde{P}} \sum_{i=1}^{T} (D_{KL}(P \parallel \tilde{P}) - D_{KL}(Q_i \parallel Q))^2 \tag{20}$$

**To prove the reasonableness of our approximation, we provide a more rigorous proof below. Our initial objective is as follows:**

$$\arg\min_{\tilde{P}} \sum_{i=1}^{T} (D_{KL}(\tilde{P} \parallel P) - D_{KL}(J||Q))^2 \tag{21}$$

By assuming $D_{KL}(Q||J) = D_{KL}(Q||Q_i) + C_J(Q_i)$, we can transform the original problem

$$\arg\min_{\tilde{P}} (D_{KL}(\tilde{P} \parallel P) - D_{KL}(J||Q))^2 \tag{22}$$

into $T$ constrained optimization problems:

$$\arg\min_{\tilde{P}} (D_{KL}(\tilde{P} \parallel P) - D_{KL}(Q_i \parallel Q) - C_J(Q_1))^2$$

$$... \tag{23}$$

$$\arg\min_{\tilde{P}} (D_{KL}(\tilde{P} \parallel P) - D_{KL}(Q_i \parallel Q) - C_J(Q_T))^2$$

After jointly optimizing them, we have:

$$\arg\min_{\tilde{P}} \sum_{i=1}^{T} (D_{KL}(\tilde{P} \parallel P) - D_{KL}(Q_i \parallel Q) - C_J(Q_i))^2$$

$$\sum_{i=1}^{T} (D_{KL}(\tilde{P} \parallel P) - D_{KL}(Q_i \parallel Q) - C_J(Q_i))^2$$

$$\leq \sum_{i=1}^{T} (D_{KL}(\tilde{P} \parallel P) - D_{KL}(Q_i \parallel Q))^2 + \sum_{i=1}^{T} C_J(Q_i))^2 \qquad (24)$$

$$= \sum_{i=1}^{T} (D_{KL}(\tilde{P} \parallel P) - KL(Q_i \parallel Q))^2 + C_{J-Q}$$

Since $C_{J-Q}$ is a constant term independent of $\tilde{P}$, we can ignore it. Finally, we solve the original problem by optimizing this upper bound. When we symmetrize the terms in the KL divergence, we can obtain a similar conclusion. Therefore, in the multi-task setting, we can solve it using the following formula: (As shown in Equation (6)):

$$\arg\min_{\tilde{P}} \sum_{i=1}^{T} \left[ (KL(P \parallel \tilde{P}) - D_{KL}(Q_i \parallel Q))^2 + (KL(\tilde{P} \parallel P) - KL(Q \parallel Q_i))^2 \right] \qquad (25)$$

# F  Pseudo Code

---

**Algorithm 1** Adaptive Logits-Arithmetic

---

**Require:** generation prompt $X$, number of tokens to generate $N$, Large model $\theta^L$, domain number $T$, expert small models $\{\theta_t^S\}_{t=1}^{T}$, $M^L$, $M^S$, and $M_t^S$ represent the logits outputs of the large model, small model, and $t$-th domain-specific small models. $P$, $Q$, and $Q_t$ represent the outputs distribution of the large model, small model, and $t$-th domain-specific small models.

1:  $k \leftarrow len(X) - 1$
2:  $m \leftarrow \infty$
3:  $\alpha \leftarrow []$
4:  **while** $len(X) < N$ **and** $x_{k-1} \neq$ [EOS] **do**

5:     **for** each domain $t$ **in** $T$ **do**
6:        Get domain expert logits for $x_k$ from $\theta_t^S$ as $M_t^S(x_k|x_{<k})$
7:     **end for**

8:     # For multitask scenario, perform $\alpha$ search only for one task, for a total of $T$ times.
9:     **for search** $\alpha' \in \mathbb{R}^T$ **do**                       ▷ such as [0.0,2.0], step is 0.1
10:        $\tilde{P}(x_k \mid x_{<k}) \leftarrow$ softmax $\left[ M^L(x_k|x_{<k}) + \sum_{t=1}^{T} \alpha'_t \cdot (M_t^S(x_k|x_{<k}) - M^S(x_k|x_{<k})) \right]$
11:        $\mathcal{L} \leftarrow \sum_{t=1}^{T} \left( KL(\tilde{P}||P) - KL(Q_t||Q) \right)^2 + \left( KL(P||\tilde{P}) - KL(Q||Q_t) \right)^2$
12:        **if** $\mathcal{L} < m$ **then**
13:           $\alpha \leftarrow \alpha'$
14:           $m \leftarrow \mathcal{L}$
15:        **end if**
16:     **end for**

17:     Calculate next token distribution for the large model as $\tilde{P} \leftarrow$ softmax $\left[ M^L(x_k|x_{<k}) + \sum_{t=1}^{T} \alpha_t \cdot (M_t^S(x_k|x_{<k}) - M^S(x_k|x_{<k})) \right]$
18:     Sample the next token $x_k \sim \tilde{P}(x_k|x_{<k})$
19:     $X \leftarrow \{X; x_k\}$
20:     $k \leftarrow k + 1$
21: **end while**
22: **return** generated text $X$

---

Our algorithm is outlined in pseudo-code in Algorithm 1, we search for each task's $\alpha$ with a step of 0.1.

## G    Dataset Details

- **GSM8K** [8], a dataset of 8.5K high-quality linguistically diverse grade school math word problems created by human problem writers. We evaluate using exact match (EM.).
- **TruthfulQA** [32], a benchmark to measure whether a language model is truthful in generating answers to questions. There is no training set for this dataset. We evaluate using multiple-choice accuracy (Acc.).
- **TriviaQA** [21], a reading comprehension dataset containing over 650K question-answer-evidence triples. We evaluate using exact match (EM.).
- **MMLU** [13], a benchmark designed to evaluate the capabilities of language models. It comprises approximately 16,000 multiple-choice questions across 57 tasks. We evaluate using multiple-choice accuracy (Acc.).
- **CNN-DailyMail (CNN/DM)** [47], a dataset for text summarization. We evaluate using Rouge2.

The GSM8K, MMLU datasets are licensed under the MIT License. The TruthfulQA, TriviaQA, CNN-Dailymail datasets are licensed under the Apache-2.0 License. Thus, these datasets are available for non-commercial research purposes.

## H    Case study

As shown in Table 4, our method achieves the correct output of 18 when targeting a single task, such as the GSM8K problem. Additionally, it can be observed that the model's output contains formatted information, e.g., "####" which matches the format learned in in-context learning. This indicates that our method achieves the ability of in-context learning at the logit level.

For unseen tasks like MMLU's Abstract Algebra and Clinical Knowledge, our model generates correct answers by leveraging multi-expert knowledge transfer.

## I    Limitations and Future Works

Our approach shares the general limitations of existing logit-arithmetic-based methods: (1) Since our method operates at the decoding step, this architecture mainly tests its ability in generative models, without exploring its effectiveness in discriminative models; (2) Involving multiple models in the decoding step can slow down inference compared to inference with models of similar magnitude.

In the future, we will explore more efficient methods of logit arithmetic and investigate their potential application in other architectural models.

## J    Broader Impacts

Our approach enables weak-to-strong knowledge transfer without accessing the model's parameters or expert-learning data. This is advantageous for protecting the privacy of expert data. Additionally, this data-free fine-tuning paradigm significantly reduces the cost of private deployment while enhancing data security, thus positively impacting societal data security. Our method does not exhibit any significant negative societal impacts.

Table 4: Case Study

| |
| --- |
| **GSM8K** |
| **Q:** Janetž2019s ducks lay 16 eggs per day. She eats three for breakfast every morning and bakes muffins for her friends every day with four. She sells the remainder at the farmers' market daily for $2 per fresh duck egg. How much in dollars does she make every day at the farmers' market? |
| **LLAMA2-13B:**
n
nAnswer: |
| **LLAMA2-13B + In-Context Learning (5-shot):** Janet eats 3 eggs for breakfast every morning.
She bakes muffins for her friends every day with 4 eggs.
She sells the remainder at the farmers' market daily for $2 per fresh duck egg.
So she makes $2 x 13 = $«2*13=26»26 every day at the farmers' market.
#### 26 |
| **Ours (LLAMA2-13B + GSM8K expert):** Janet eats 3 eggs for breakfast and bakes 4 eggs into muffins for a total of 3 + 4 = «3+4=7»7 eggs per day.
She has 16 - 7 = «16-7=9»9 eggs left to sell at the farmers' market.
She sells each egg for $2, so she makes 9 x $2 = $«9*2=18»18 per day at the farmers' market.
#### 18 |
| **MMLU: Abstract Algebra** |
| **Q:** Statement 1 ∣ Every group of order 42 has a normal subgroup of order 7. Statement 2 ∣ Every group of order 42 has a normal subgroup of order 8.
A. True, True
B. False, False
C. True, False
D. False, True ? |
| **LLAMA2-13B:** D. False, True
Question: Statement 1 ∣ Every group of order 42 has a normal subgroup of order 7.
Statement 2 ∣ Every group of order 4 |
| **Ours LLAMA2-13B + GSM8K expert + TruthfulQA expert + TriviaQA expert + CNN DM expert:** C. True, False . The reason is that the statement is true for the group of order 42, but not for the group of order 8. |
| **MMLU: Clinical Knowledge** |
| **Q:** Wound drainage is described as serous if:
A. it contains red blood cells.
B. it is thick and yellow or green.
C. it contains microorganisms.
D. it is clear or slightly yellow. |
| **LLAMA2-13B:** D
Question: Which of the following is not a characteristic of a normal blood film? |
| **LLAMA2-13B + In-Context Learning (5-shot):** D. it is clear or slightly yellow. |
| **Ours (LLAMA2-13B + GSM8K expert + TruthfulQA expert + TriviaQA expert + CNN DM expert):** D. it is clear or slightly yellow. |

Table 5: Case Study(token routing)

| |
| --- |
| **GSM8K** & **TruthfulQA** & **Common** |
| **Q:** A microwave oven is connected to an outlet, 120 V, and draws a current of 2 amps. At what rate is energy being used by the microwave oven? |
| **LLAMA2-7B:** The rate at which energy is being used by the microwave oven can be calculated by multiplying the voltage (120 V) by the current (2 amps). This gives us a power rating of 120 V x 2 A = 240 watts.\n#### 240 |

