# OpenReview forum: "On Giant's Shoulders: Effortless Weak to Strong by Dynamic Logits Fusion"
_NeurIPS.cc/2024/Conference — NeurIPS 2024 poster_

### Official Review · Reviewer_Ttwe · 2024-07-12

**Soundness:** 3
**Presentation:** 2
**Contribution:** 2
**Rating:** 4
**Confidence:** 3

**Summary:**

The article discusses existing weak-to-strong methods, noting that current approaches typically use a static knowledge transfer ratio and a single small model to convey complex knowledge, which results in suboptimal performance. Consequently, the article proposes a dynamic logit fusion method that employs a series of task-specific small models and adaptively allocates weights among these models at each decoding step. The weights are learned by optimizing a problem constrained by Kullback-Leibler divergence. The article conducts experiments on various benchmarks, including both multi-task and single-task scenarios.

**Strengths:**

1. The article reevaluates existing logit arithmetic methods, highlighting the significant impact of fusion weights and the limitations of a single small model on test performance.

2. By using constrained optimization, the article autonomously learns fusion weights, thereby approximating the computationally intensive results of fine-tuning large foundational models.

3. Experiments were conducted to validate the proposed method, demonstrating notable improvements in performance, generalization capability, and robustness.

**Weaknesses:**

1. In Section 3.2, the proposed method is based on an assumption that lacks clear supporting evidence. Does this somewhat undermine the theoretical foundation of the algorithm?

2. In the multi-task experiments, the results for CNN/DM do not demonstrate the method's superiority. Additionally, in the experiments on unseen tasks, the method does not show significant improvement.

3. In the experiments, all evaluations were conducted in a 0-shot setting. How would the evaluation results change if a 5-shot experiment were conducted?

**Questions:**

1. In the multi-task setting, which 7B model is used as the expert to implement your algorithm on the 13B model? I am very confused. If you are using different sets of experts to operate on the 13B model, isn't this weak-to-strong? Because the parameters of multiple 7B models exceed 13B.
2. There seems to be a typo in formula (9) in Appendix B, it seems to be $\propto$ rather than $=$.

**Limitations:**

The authors have adequately addressed the limitations.

---

> ### Author Rebuttal · Authors · 2024-08-07
>
> > Q1: In the multi-task setting, which 7B model is used as the expert to implement your algorithm on the 13B model? I am very confused. If you are using different sets of experts to operate on the 13B model, isn't this weak-to-strong? Because the parameters of multiple 7B models exceed 13B
>
> In the multi-task setting, we used all the experts for the seen tasks, i.e., four experts fine-tuned on each task (7B or 1.1B).
>
> It needs to be clarified that our "weak to strong" approach aims to use weak model supervision to elicit the full capabilities of a much stronger model. In our setting, we use small fine-tuned models (weak models, e.g., 7B or 1.1B) that have transferred downstream task-specific knowledge to enhance a large model (strong model, e.g., 13B) without downstream knowledge. In our setting, "weak" or "strong" indicates the capability upper bound of a model, which is commonly enhanced by scaling model size. However, the capabilities of multiple small models are constrained by their model size and are generally weaker than the large ones, as shown by the performance of the 7B multi-task expert on unseen tasks in Table 2 (35.82<51.25).
>
> Additionally, fine-tuning a large model is significantly more expensive than fine-tuning a small model, requiring more advanced hardware and more training time. In contrast, our weak-to-strong paradigm only needs to fine-tune the small models.
>
>
> > Q2: There seems to be a typo in formula (9) in Appendix B, it seems to be $\propto$ rather than $=$.
>
> Thanks for your valuable advice. We will carefully revise our paper based on your suggestion.
>
>
> > Q3:In Section 3.2, the proposed method is based on an assumption that lacks clear supporting evidence. Does this somewhat undermine the theoretical foundation of the algorithm
>
> Our work is based on verified theories [1, 2, 3]. First, using the shift of the fine-tuned model to accomplish our domain adaptation task is reasonable. To demonstrate our optimization process, we can view the fine-tuning procedure as reinforcement learning (RL) with a KL-divergence constraint preventing divergence from a reference model.
>
>   *According to the theory presented in the DPO[1]: Theorem 1. Under mild assumptions, all reward classes consistent with the Plackett-Luce (and Bradley-Terry in particular) models can be represented with the reparameterization* $r(x, y) = \beta \log \frac{\pi(y|x)}{\pi_{ref}(y|x)}$ *for some model* $\pi(y|x)$ *and a given reference model* $\pi_{ref}(y|x)$.
>
>   Meanwhile, According to [1,2] the optimal solution to the KL-constrained reward maximization objective is given by:
> $$
> \begin{align}
>   \pi_r(y|x)=\frac{1}{Z(x)}\pi_{ref}(y|x)exp(\frac{1}{\beta}r(x,y))\\\\
>   where\quad Z(x)=\sum_y \pi_{ref}(y|x)exp(\frac{1}{\beta}r(x,y))
> \end{align}
> $$
>
>
>   Combining the above theory, we can derive the following equation:
>
>   $\pi_{r}(y|x)=\frac{1}{Z(x)}\pi_{ref}(y|x)exp(\log \frac{\pi_r(y|x)}{\pi_{ref}(y|x)})$
>
>   Since any language model can be viewed as the solution to KL-constrained RL with a constraint to the pre-trained model[1], this equation is applicable to fine-tuning scenarios. We can replace
>   $\pi$ in the parentheses with the small model's $\pi$, resulting in the following equation:
>
>
> $\pi_{L-ft}(y|x)=\frac{1}{Z(x)}\pi_{L-pt}(y|x)exp(\log \frac{\pi_{S-ft}(y|x)}{\pi_{S-pt}(y|x)})$
>
> It can be seen that, based on the theory from previous work, it is reasonable to assume that the shifts between models are consistent for knowledge transfer. This is consistent with the form of the proof in Appendix B.
>
> Compared to global static transfer, we adjust an appropriate shift at each decoding step to achieve better transfer results. KL divergence[1, 2, 3] is commonly used to describe the distance between distributions (as shown in section 3.1), making it more suitable for representing the shift between two distributions. We use KL divergence as a distance function to represent the above shift, converting it into a KL-constrained problem, dynamically controlling the knowledge transfer by constraining each decoding step. Meanwhile, the squared error is commonly used in various regression prediction approximations[3], and it is easy to solve, making it well-suited for our setup. Additionally, as shown in the "Supplementary Proof for the Fusion of Multiple SLMs Scenario" in the Global Rebuttal, due to the geometric properties and inequality characteristics of squared error, our method can extend more smoothly to scenarios involving multiple experts when using squared error.
>
> - [1] Direct preference optimization: Your language model is secretly a reward model (NIPS2023)
> - [2] RL with KL penalties is better viewed as Bayesian inference (EMNLP2022)
> - [3] Learning Theory for Distribution Regression (JMLR2016)
>
> > Q4:In the multi-task experiments, the results for CNN/DM and unseen tasks do not show significant improvement
>
> Actually, in our multi-task experiments, our method achieved a 16% improvement on CNN/DM compared to the 13B model (8.94->10.52). It is noticeable that after using multi-task tuning on the 13B model, the performance on unseen tasks even decreased (51.28->50.58). This indicates that unseen tasks are more challenging and that overfitting can occur when training on seen tasks. In contrast, our method shows an improvement on unseen tasks (51.28->51.31), demonstrating that our approach not only provides significant enhancements on seen tasks but also helps mitigate overfitting within the domain.
>
> > Q5: How would the evaluation results change if a 5-shot experiment were conducted?
>
>  Actually, we have conducted 5-shot experiments. As mentioned in Section 5.3, our method can be combined with in-context learning(ICL) and can also enhance its effect. We used the 5-shot approach as the ICL setting. As shown in Figure 5(a), our method combined with 5-shot ICL achieves an overall improvement of 18.3% compared to using ICL alone. This is mainly due to our method's ability to integrate the knowledge possessed by the experts.

---

> ### Author Response · Authors · 2024-08-12
>
> Dear Reviewer Ttwe:
>
> We wish to thank you again for your constructive feedback which has helped us to improve the clarity and contribution of our work. As the discussion period draws to a close, we hope our response has effectively addressed all your concerns. Your insights are invaluable to us, and we remain open to further discussion if you have any questions regarding our response.

---

> > ### Comment · Reviewer_Ttwe · 2024-08-12
> >
> > Thank you for answering my question, but I'm afraid I disagree with your response to the first question. As I understand it, if you use four 7B experts to fine-tune a 13B model, this doesn't qualify as **weak to strong**. In fact, you're using a 28B MoE model to fine-tune the 13B model. If you could clarify this point better, I would consider raising the score.

---

> > > ### Author Response · Authors · 2024-08-12
> > >
> > > We thank the reviewer for the feedback. We will address the concerns below.
> > >
> > > 1、It should be noted that our method does not require fine-tuning the 13B model. Instead, our approach involves fine-tuning multiple 7B models and then transferring the knowledge to the 13B model without the need for gradients.
> > >
> > > 2、Although we have multiple 7B models, their capability is still far from reaching the level of a 28B model. Therefore, we cannot view the entire process as a migration from 28B to 13B. As shown in the table below, the 7B-Expert Best (the best result from each dataset within the 7B-expert models) still struggles to outperform the 13B Multi-Task Tuning results, especially on unseen tasks. Other gradient-free methods (e.g., average[1], task arithmetic[2]) that do not involve training also find it difficult to combine four 7B models into a strong model equivalent to 28B, and they even fall short of surpassing the 13B model. Therefore, without training, the capability of four 7B models is weaker than that of the 13B model.
> > >
> > > |                       | seen task | unseen task |
> > > | --------------------- | --------- | ----------- |
> > > | 13B Multi-Task Tuning | **40.78** | **50.58**   |
> > > | 7B-Expert Best        | 40.02     | 46.61       |
> > >
> > > 3、In a multi-task setting, the logit arithmetic in formula (6) can be expressed as $13B+\sum_i^m  \alpha_i diff^{7B}_i=\sum_i^m (\frac{1}{m}13B+\alpha_i diff^{7B}_i)$ ，where $diff^{7B}_i$ represents the shift of the i-th expert on the 7B-base in logits. Compared to the 28B MoE, our method actually performs as a combination of multiple weak-to-strong approaches (marked by parentheses in the above formula). Our method adjusts $\alpha$ to control their combination ratio, thereby facilitating the transfer of multiple small experts, as described in the Supplementary Proof for the Fusion of Multiple SLMs Scenario in the Global Rebuttal.
> > >
> > >
> > >
> > > - [1] Model soups: averaging weights of multiple fine-tuned models improves accuracy without increasing inference time. (ICML2022)
> > >
> > > - [2] Task Arithmetic in the Tangent Space: Improved Editing of Pre-Trained Models (NIPS2023)

---

> > > > ### Comment · Reviewer_Ttwe · 2024-08-13
> > > >
> > > > Thanks for your response, I have raised the point.

---

> > > > > ### Author Response · Authors · 2024-08-13
> > > > >
> > > > > Dear Reviewer Ttwe:
> > > > >
> > > > > We thank the reviewer for their feedback and for raising the score!
> > > > >
> > > > > If you still have any unresolved concerns about our paper, we are open to further discussion. If you feel that the previously mentioned weaknesses (or questions) have been addressed and have no remaining concerns, would it be possible for you to reconsider our paper?

---

### Official Review · Reviewer_frHX · 2024-07-13

**Soundness:** 3
**Presentation:** 4
**Contribution:** 3
**Rating:** 7
**Confidence:** 4

**Summary:**

They tackle the problem of merging the logits from multiple models. To do so, they propose an objective that minimizes the squared loss of the KL between the two pairs of (student, teacher) models. This is solved via a random search.

**Strengths:**

- Paper well written and easy to follow
- Nice ablations on alpha and efficiency
- Works well in single task scenarios

**Weaknesses:**

- Using the squared error between two KL’s is not theoretically motivated (at least that I am aware of)
- More description of the optimization method in main text since it is a big part of the method
- Missing baseline in multi-task tuning setup

**Questions:**

- Where is the baseline proxy tuning for multi-task tuning in Table 2?
- From algorithm 1 in Appendix C, the optimization is done just via random search? Guessing values and storing the best? I couldn’t find the method to optimize the objective mentioned in the main paper. It should be more clearly stated in the main paper, and how it handles multitask setup.
- For the efficiency analysis, the BV can be done in 1 forward pass all in parallel, but the n parameter searches require 20 forward passes. It really is n times more compute, which is not minimal. For example, decoding 100 tokens vs decoding 1 token would be a constant in the Big O, but they are different efficiency wise.
- In eq (2), is the second term (multiplied by alpha) not normalized but in logit space?
- In fig 4, are there some special about the tokens being generated at the timesteps where $alpha$ is high?

---

> ### Author Rebuttal · Authors · 2024-08-07
>
> > Q1:Using the squared error between two KL's is not theoretically motivated (at least that I am aware of)
>
> The goal of our motivation is to enhance the constraints using KL divergence, aiming for the shift of the fine-tuned large model to be equal to the shift of the fine-tuned small models in each decoding step. Compared to global static transfer, we adjust an appropriate shift at each decoding step to achieve better transfer results. We use KL divergence as a distance function to measure this "shift," (as shown in section 3.1) and the squared error between KL divergences helps us align these two shifts. Squared error is commonly used in various regression prediction approximations[1], and it is easy to solve, making it well-suited for our setting. Additionally, as shown in the "Supplementary Proof for the Fusion of Multiple SLMs Scenario" in the Global Rebuttal, due to the geometric properties and inequality characteristics of squared error, our method can extend more smoothly to scenarios involving multiple experts when using squared error.
>
> - [1] Learning Theory for Distribution Regression (JMLR2016)
>
>
> > Q2:More description of the optimization method in main text since it is a big part of the method.
>
>   Thank you for your reminder. We will elaborate on this section in the next version. Our optimized method performs a linear search for $\alpha$ and multiple Logi Arithmetic operations after obtaining the logits from all models to find the optimal situation described in Eq(6). During the search, we start from 0 and increment by 0.1 each time until we reach 2, resulting in 20 searches.
>   We perform this search at each decoding step, so, as shown in Figures 3 and 4, the $\alpha$ varies for each decoding step.
>
> In the multi-task setting, directly searching for every expert results in exponential complexity. For practical use, we accelerate this process by using only one small expert at each decoding step, thereby reducing the exponential complexity of the search process to linear complexity. Experiments have shown that this approach also yields good results (as shown in Table 2).
>
>
> > Q3: Where is the baseline proxy tuning for multi-task tuning in Table 2?
>
> It is worth noting that Proxy Tuning, due to its lack of prior assumptions in a multi-task setting and the difficulty in presetting the transfer proportions for multiple experts, is not capable of handling multi-task scenarios. In contrast, our method can dynamically adjust the transfer proportions of expert knowledge, making it naturally suitable for multi-task settings.
>
> To better demonstrate the effectiveness of our method, we further compared it with our method with static $\alpha$ in a multi-task setting. For the 4 seen tasks in our experiment, we set the corresponding expert coefficient to 0.25 ($\alpha=1/4$, assigning the same proportion to each seen task expert). In the table below, it can be seen that our method, which dynamically adjusts the coefficients, significantly outperforms the static setting.
>
> |  | Seen Task | Unseen Task | Avg. |
> | --- | --- | --- | --- |
> | Ours (0.25 static) | 22.02 | 46.04 | 34.03 |
> | Ours | **27.53** | **51.31** | **39.42** |
>
>
> > Q4:For the efficiency analysis, the BV can be done in 1 forward pass all in parallel, but the n parameter searches require 20 forward passes. It really is n times more compute, which is not minimal. For example, decoding 100 tokens vs decoding 1 token would be a constant in the BigO, but they are different efficiency wise.
>
>  Actually, our method only performs one forward pass when doing logit arithmetic. As analyzed in the "Complementary to Efficiency Analysis" section of the Global Rebuttal, the $nBV$ term represents $n$ quick logit arithmetic operations to obtain the final logits, which only requires one forward pass and not $n$ forward passes. So overall, the time consumption is almost the same as that of the static method.
>
>
> > Q5:In eq (2), is the second term (multiplied by alpha) not normalized but in logit space?
>
>   The second term is not normalized. Normalization is performed after the entire logit arithmetic calculation is completed.
>
>
> > Q6:In fig 4, are there some special about the tokens being generated at the timesteps where alpha is high?
>
>   When $\alpha$ is high, the confidence in the logits generated by a specific expert will be higher, leaning towards the tokens that this expert is more certain about it at the moment. For example, for gsm8k, a high $\alpha$ will tend to generate mathematical symbols.
>
>   For the following question:
> {"question": "A pen costs as much as a pencil and eraser combined. A pencil costs \\$1.20 and an eraser costs \\$ 0.30. How much will 8 pens cost?"}
>
> The answers obtained from our method are as follows (**bold** indicates $\alpha$ is the upper bound, and `red` indicates $\alpha$ to the lower bound):
>
> {Ours: " **8** pencils will cost **8** * `$`1.2**0** = `$<<`8\***1.2=9.60>>9.60**. 8 **erasers** `will cost` **8 \*** `$`**0**.30 = **\$<<8*0.30=2.40>>2.40.** `Thus`, **8** `pens will` cost \$9.**60** **+** \$**2.40** `= $<<`**9.6+2.4=12>>12**."}
>
>   As can be seen, when $\alpha$ is at the upper bound, the response leans more towards mathematical reasoning; when $\alpha$ is at the lower bound, the response tends to be more of a normal statement or information about the question.

---

> > ### Comment · Reviewer_frHX · 2024-08-12
> >
> > Thanks for the response. I keep my score as is.

---

### Official Review · Reviewer_YRcj · 2024-07-13

**Soundness:** 3
**Presentation:** 2
**Contribution:** 3
**Rating:** 6
**Confidence:** 4

**Summary:**

The paper studies the problem of adapting large general language models via smaller expert language models fine-tuned on specific tasks. Prior work proposed the idea of mixing logits between a large model and the differencei in logits pre- and post- finetuning of a small model. The authors take this idea a step further and compute the mixing weights adaptively per-token, leading to better results.

**Strengths:**

1. The method is very simple: the authors match tune the weights to match the KL divergence between the small model before and after fine-tuning, for each token.

2. The experiments are comprehensive with 5 tasks and 2 small models (1.1B and 7B). The authors also consider both single-task and multi-task scenarios.

3. The results are good across all tasks, the proposed method outperforms proxy-tuning as well as full fine-tuning on the smaller model predictions, and recovers a large fraction of the ceiling performance achieved by directly finetuning the large model on ground truth.

4. There are several ablations and understanding experiments in Section 5.

**Weaknesses:**

1. It is not intuitively obvious to me why matching the KL divergence is the right objective. Could the authors please provide some intuition? I imagine it is something like this: when the small model updates significantly for some token, we want the large model to also udpate significantly. That seems reasonable, but probably doesn't always work well: if the small model is unaware of some fact or makes an arithmetic mistake, it may need to update significantly on the corresponding tokens, while a large model does not need to update.

2. The presentation is not always very clear. For example, in Eq. (5) it is not clear to me what the authors mean by the joint distribution of $Q_1, Q_2, \ldots, Q_T$. How can we compute KL between this joint and $Q$?

3. As the authors mention, the proposed method is 2.5 times slower at inference time compared to standard sampling from the same model.

**Questions:**

1.1 and 1.2: See weaknesses 1 and 2.

2. What exactly do you mean by supervised instruction tuning: what are the smaller models fine-tuned on? Are these chain-of-thoughts for solving the task, e.g. GSM8k? Where do they come from?

3. In Figure 3, qualitatively, what do the tokens (decoding steps) where we set the weight to the lower bound correspond to, and same for the upper bound? Are they qualitatively different?

4. In Figure 3, why is the lower bound 0.8 and the upper bound 1.5? Are these tunable parameters? 0.8 seems quite high, shouldn't we want to set the lower bound to 0?

**Limitations:**

Limitations are adequately discussed.

---

> ### Author Rebuttal · Authors · 2024-08-07
>
> > Q1: It is not intuitively obvious to me why matching the KL divergence is the right objective. Could the authors please provide some intuition? I imagine it is something like this: when the small model updates significantly for some token, we want the large model to also udpate significantly. That seems reasonable, but probably doesn't always work well: if the small model is unaware of some fact or makes an arithmetic mistake, it may need to update significantly on the corresponding tokens, while a large model does not need to update.
>
> Our method has already considered this situation, which presents a challenge for static methods. Our goal is to control the shift of the fine-tuned model to be the same, and KL divergence is commonly used to describe the distance between distributions, making it more suitable for representing the shift between two distributions.
>
> Using a static shift to match on a sentence level can indeed result in the incorrect transfer of some erroneous knowledge. Therefore, our method refines this process to each decoding step, allowing dynamic adjustment of knowledge transfer intensity to mitigate this issue. Notably, our method ultimately overlays the transferred knowledge onto the logits of the large model. **This means that when the small model generates errors, our method dynamically adjusts based on the large model's capabilities. When the large model can solve the problem independently, it will retain more of its own abilities.**
>
> As shown in the results of Table 1, our method significantly improves the model's performance even when the effect of the 1.1B finetuned model is much lower than that of the 13B base model. For instance, on MMLU, the 1.1B model improves from 37.26 to 48.32. Compared to the static method's improvement from 37.26 to 39.88, **our method does not fully trust the capabilities of the small model but rather retains more of the large model's abilities to mitigate this issue.**
>
>
>
> > Q2: How can we compute KL between this $Q_1...Q_T$ and $Q$?
>
>
> The term "joint distribution" in our paper refers to the fusion distribution obtained by combining the outputs of a series of smaller models.
>
>  As shown in the "Global Rebuttal" section "Supplementary Proof for the Fusion of Multiple SLMs Scenario," we transform the problem of approximating $J$ into a centroid problem (i.e., optimizing the upper bound of $KL(J||Q)$). Therefore, we can use equation (6) in the paper to calculate this KL divergence.
>
> > Q3: The proposed method is 2.5 times slower at inference time compared to standard sampling from the same model.
>
> It should be noted that the 2.5x slower inference speed is compared to the 13B FFT (full fine-tuning). **However, our method does not require fine-tuning the large model (e.g., 13B), allowing it to benefit from smaller expert models, resulting in much lower hardware requirements.** As shown in Table 3, the time required for 13B FFT is 1176s, while the time required for 7B FFT or 1.1B FFT is only 588s or 128s, significantly less than the time required for 13B FFT.  Additionally, our method can leverage many pre-existing small expert models from Huggingface, further reducing training time.
>
> Furthermore, as noted in the "Complementary to Efficiency Analysis" section of our Global Rebuttal, the time consumed by our method is almost identical to that of the static method, while our method performs significantly better.
>
>
> > Q4: Supervised instruction tuning details.
>
> For each task, we used the official dataset and trained our small model on the official training set. We conducted supervised instruction tuning without using chain-of-thoughts. When constructing the prompt, we used simple instructions for concatenation. For example, for gsm8k: "Question: " + [question] + "\\nAnswer:". For CNN/DM: [article] + "\\n\\nSummarize the above article:".
>
>
> > Q5:In Figure 3, what do the tokens where we set the weight to the lower bound correspond to, and what about the upper bound? Are they qualitatively different?
>
> When $\alpha$ is high, the confidence in the logits generated by a specific expert will be higher, leaning towards the tokens that this expert is more certain about at the moment. For example, for gsm8k, a high $\alpha$ will tend to generate mathematical symbols.
>
> For the following question: {"question": "A pen costs as much as a pencil and eraser combined. A pencil costs \\$1.20 and an eraser costs \\$ 0.30. How much will 8 pens cost?"}
>
> The answers obtained from our method are as follows (**bold** indicates $\alpha$ is the upper bound, and `red` indicates $\alpha$ to the lower bound):
>
> {Ours: " **8** pencils will cost **8** * `$`1.2**0** = `$<<`8\***1.2=9.60>>9.60**. 8 **erasers** `will cost` **8 \*** `$`**0**.30 = **\$<<8*0.30=2.40>>2.40.** `Thus`, **8** `pens will` cost \$9.**60** **+** \$**2.40** `= $<<`**9.6+2.4=12>>12**."}
>
> **As can be seen, when $\alpha$ is at the upper bound, the response
> leans more towards mathematical reasoning; when $\alpha$ is at the
> lower bound, the response tends to be more of a normal statement or
> information about the question.**
>
> > Q6:In Figure 3, why is the lower bound 0.8 and the upper bound 1.5? Are these tunable parameters? 0.8 seems quite high, shouldn't we want to set the lower bound to 0?
>
>   Sorry for the confusion, Figure 3 can indeed be misleading. In our experiments, $\alpha$ was searched from 0 to 2.0. The values 0.8 and 1.5 in Figure 3 represent the minimum and maximum values obtained during the optimization process. In the GSM8K task, due to the large model's inherent capability bias, the overall trust in the expert knowledge is relatively high, resulting in higher values obtained during the optimization process. We will improve the depiction of the figure in the next
>   version.

---

> ### Author Response · Authors · 2024-08-12
>
> Dear Reviewer YRcj:
>
> We wish to thank you again for your constructive feedback which has helped us to improve the clarity and contribution of our work. As the discussion period draws to a close, we hope our response has effectively addressed all your concerns. Your insights are invaluable to us, and we remain open to further discussion if you have any questions regarding our response.

---

### Official Review · Reviewer_XMhg · 2024-07-13

**Soundness:** 2
**Presentation:** 3
**Contribution:** 2
**Rating:** 5
**Confidence:** 4

**Summary:**

This paper focuses on the weak-to-strong generalization paradigm where the goal is to transfer knowledge from a small language model to larger one. The method they study is the one proposed by Mitchell et al. [1]: they use log probability algebra to combine the logits of the large model, the ones of a small model and the ones of a small model that has been finetuned. This combination involves a parameter $\alpha$ that controls the contribution of the small model. The main contributions of this paper is to point to the limitations of using a static $\alpha$ and to propose a method to adaptively learn such $alpha$. Their method consists in optimizing an objective based on the KL divergence and they show that their approach is consistently better than using a static alpha across a wide range of downstream tasks.




[1] Mitchell, Eric, et al. "An emulator for fine-tuning large language models using small language models." arXiv preprint arXiv:2310.12962 (2023).

**Strengths:**

I find the paper is well written and the methodology well presented. The authors did a great job at presenting the problem, the limitations of the current methods to solve the problem and their method. They also did a good job at presenting their experiments.

**Weaknesses:**

Overall, my main concern is that I find the contribution limited and I have some doubts about the method. Here is a detailed list of my concerns:

- **Computational feasability**: I think that the authors should be more transparent in the computational cost of their procedure. Solving the optimizaiton problem at every decoding step may be very expensive and it is not clear to me that when one needs to do many generations, a finetuned model with LoRA on the large model is cheaper than the procedure proposed by the authors. Also, is it important to update alpha at each decoding step? Can't one get a more efficient procedure by updating it only every 100 tokens or so?

- **Not enough clear to me that the method does much better than the static $\alpha$**: when I see the barplot of figure 2, it seems like $\alpha=1$ is a bit below than the learnt $\alpha$ but the gap is not huge.

- **Theoretical justification for the optimization problem?**: So if I understand correctly, the authors objective function is to say "I want the distance between the predictions of $\tilde{P}$ and the large model is the same as the distance between the predictions of the finetuned small model and the small model".   This looks like a reasonable belief. However, it would have been nice to have a theoretical justification. For instance, when you do RL finetuning, if you solve the problem exactly, the distribution you end up generating from ends up being p_bayes(generation) * exp(reward model you are training on) (usually done with ppo, dpo). When you take the logs, you get log(p) = log(p_bayes) + reward model.  Then you can estimate the reward model by taking the difference of logits for any model scale and in this case the intuition of the authors make sense to me and it is principled. However, in the standard finetuning case, when the authors apply this intuition at the level of tokens, it is not clear to me why it should work.

- **Scaling experiments for studying the approach**: I know that the Llama suite starts at 7B but it would have been nice to study the behavior of the method with smaller models than 7b.  Understanding how robust the method is by varying the gap between the weak and strong models is important. There are maybe chances that the learnt $\alpha$ approach shows bigger  gaps with respect to the static $\alpha$ when the gap between the weak and strong models is large.

**Questions:**

I would appreciate that the authors address the concerns I have regarding the theoretical justification of their optimization problem, the computational cost of their approach.

**Limitations:**

I think that the authors didn't clearly state the limitiations of their approach, which is regretful.

---

> ### Author Rebuttal · Authors · 2024-08-07
>
> > Q1:Solving the optimization problem at every decoding step is expensive?
>
> As we analyzed in the "Complementary to Efficiency Analysis" section of the Global Rebuttal, our method only adds the term "$nBV$" compared to the static method. Optimizing $n$ times ($n \le 20$)  during each forward pass is negligible compared to the overall forward pass time. In the experiments, as shown in Table 3, our method is only 0.008s slower per data point on average compared to the static method.
>
>
> > Q2: A finetuned model with LoRA on the large model is cheaper?
>
> SFT with LoRA still requires forwarding the full large model thousands
>   of times (e.g., 13B) to tune LoRA during training. **In contrast, our
>   method does not require fine-tuning the large model at all, benefiting
>   from transferring from smaller models (7B or 1.1B) or using
>   pre-existing models (e.g., from Hugging Face).** As shown in Table 3, the time for LoRA tuning a 13B model is 836 seconds, while the time for fully fine-tuning a 7B or 1.1B model is 588s or 128s, respectively, which is significantly less than the time required to fine-tune the 13B model.  LoRA tuning on the smaller models further reduces the time to 364s (7B) and 128s (1.1B). This demonstrates that fine-tuning smaller models requires less hardware resources and less time. Furthermore, utilizing pre-trained smaller models from Hugging Face allows efficient transfer without extensive training. Therefore, our method is cheaper compared to fine-tune a large model.
>
> > Q3:Updating $\alpha$ every 100 tokens is more efficient?
>
> As we analyzed in the "Complementary to Efficiency Analysis" section of the Global Rebuttal, our method and the static method have almost the same time consumption per data point on average. As shown in the below table, reducing the update frequency of $\alpha$ may negatively impact the final result. Therefore, we ultimately chose to optimize $\alpha$ at each step.
>
> |update step|1|100|$+\infty$|
> |-|-|-|-|
> | GSM8K| 39.34(0.166s/per sample) | 37.84(0.159s/per sample) | 37.68(0.158s/per sample) |
>
>
>
> > Q4: Not enough clear to me that the method does much better than the static $\alpha$: when I see the barplot of figure 2, it seems like $\alpha=1$ is a bit below than the learnt but the gap is not huge.
>
> Actually, our method outperforms the static setting of $\alpha=1.0$ on single tasks, with improvements of 4.4%, 0.9%, 8.1%, 6.5%, and 1.6% on GSM8K, TruthfulQA, TriviaQA, CNN/DM, and MMLU, respectively. Our method has a significant advantage over the static method, with an average improvement of 4.3%.
>
> To better demonstrate the effectiveness of our method, we further compared it with our method with static $\alpha$ in a multi-task setting. For the 4 seen tasks in our experiment, we set the corresponding expert coefficient to 0.25 ($\alpha=1/4$, assigning the same proportion to each seen task expert). In the table below, it can be seen that our method, which dynamically adjusts the coefficients, significantly outperforms the static setting.
>
>
> |  | Seen Task | Unseen Task | Avg. |
> | --- | --- | --- | --- |
> | Ours (0.25 static) | 22.02 | 46.04 | 34.03 |
> | Ours | **27.53** | **51.31** | **39.42** |
>
>
> > Q5: The theoretical justification for the optimization problem?: It's not clear to me why it should work in standard finetuning case?
>
> To demonstrate our optimization process, we can view the fine-tuning procedure as reinforcement learning (RL) with a KL-divergence constraint preventing divergence from a reference model.
>
> *According to the theory presented in the DPO[1]: Theorem 1. Under mild assumptions, all reward classes consistent with the Plackett-Luce (and Bradley-Terry in particular) models can be represented with the reparameterization* $r(x, y) = \beta \log \frac{\pi(y|x)}{\pi_{ref}(y|x)}$ *for some model* $\pi(y|x)$ *and a given reference model* $\pi_{ref}(y|x)$.
>
> Meanwhile, According to [1,2] the optimal solution to the KL-constrained reward maximization objective is given by:
>
> $$
> \begin{align}
>     \pi_r(y|x)=\frac{1}{Z(x)}\pi_{ref}(y|x)exp(\frac{1}{\beta}r(x,y))\\\\
>   where\quad Z(x)=\sum_y \pi_{ref}(y|x)exp(\frac{1}{\beta}r(x,y))
> \end{align}
> $$
>
>
>   Combining the above theory, we can derive the following equation:
>
>   $\pi_{r}(y|x)=\frac{1}{Z(x)}\pi_{ref}(y|x)exp(\log \frac{\pi_r(y|x)}{\pi_{ref}(y|x)})$
>
>   Since any language model can be viewed as the solution to KL-constrained RL with a constraint to the pre-trained model[1], this equation is applicable to fine-tuning scenarios. We can replace $\pi$ in the parentheses with the small model's $\pi$, resulting in the following equation:
>
> $\pi_{L-ft}(y|x)=\frac{1}{Z(x)}\pi_{L-pt}(y|x)exp(\log \frac{\pi_{S-ft}(y|x)}{\pi_{S-pt}(y|x)})$
>
> - [1] Direct preference optimization: Your language model is secretly a reward model (NIPS2023)
> - [2] RL with KL penalties is better viewed as Bayesian inference (EMNLP2022)
>
>
> > Q6 Scaling experiments for studying the approach
>
> Actually, we have chosen TinyLlama1.1B in our experiments.
> In the table below, we show the improvements of our model compared to the static method on 1.1B and 7B models in single-task experiments. The numbers in the table represent the results of the static method, with the relative improvements of our method shown in parentheses. It can be noted that when the gap between the weak model and the strong model is larger, our method indeed better adjusts the capability of expert knowledge transfer.
>
> |  | GSM8K | TruthfulQA | TriviaQA | CNN/DM | MMLU | Avg. |
> | --- | --- | --- | --- | --- | --- | --- |
> | from 1.1B | 16.91(8.0%$\uparrow$) | 31.48(17.7%$\uparrow$) | 48.74(10.4%$\uparrow$) |13.23(9.4%$\uparrow$) | 39.88(21.16%$\uparrow$) | 31.74(9.8%$\uparrow$)
> | from 7B | 37.68(4.4%$\uparrow$) | 61.02(0.9%$\uparrow$) | 52.81(8.1%$\uparrow$) |14.37(6.5%$\uparrow$) | 56.24(1.6%$\uparrow$) | 44.43(3.7%$\uparrow$) |

---

> ### Author Response · Authors · 2024-08-12
>
> Dear Reviewer XMhg:
>
> We wish to thank you again for your constructive feedback which has helped us to improve the clarity and contribution of our work. As the discussion period draws to a close, we hope our response has effectively addressed all your concerns. Your insights are invaluable to us, and we remain open to further discussion if you have any questions regarding our response.

---

> ### Comment · Reviewer_XMhg · 2024-08-13
> **Post-rebuttal response**
>
> I thank the reviewers for their rebuttal and I appreciated their theoretical derivation. I believe that this important piece to be added to the paper. I hope they will do it. i increase my score by one point.

---

> > ### Author Response · Authors · 2024-08-13
> >
> > We thank the reviewer for raising the score, and we will add this piece in our future versions.

---

### Author Rebuttal · Authors · 2024-08-07

## Global Rebuttal

Dear reviewers,

We much appreciate for your acknowledgment of our work and helpful, insightful comments. Following the reviewers' suggestions, we have carefully revised the paper and conducted a series of new experiments to address the reviewers' concerns.

The below contains a rebuttal for remarks that are common to most reviewers.

### Complementary to Efficiency Analysis (for XMhg, YRcj, frHX)
In section 5.2 of our paper, we conducted an Efficiency Analysis of logit Arithmetic. To better illustrate our efficiency, we further analyze the overall efficiency of our method here.

Overall, **our method has a similar time complexity to the static method**.  Given: current sequence length $s$, large model dimension $h_L$, small model dimension $h_S$, number of layers in the large model $L_1$, number of layers in the small model $L_2$, batch size $B$, vocabulary size $V$, number of searches per decoding step $n$. Assume the FLOPs for a single forward pass of the large model and the small model are $FLOPs_L$ and $FLOPs_S$, respectively. The FLOPs can be calculated as: $FLOPs_L=L_1*(12Bsh_L^2+2Bs^2h_L)+Bsh_LV$ ,and $FLOPs_S=L_2*(12Bsh_S^2+2Bs^2h_S)+Bsh_SV$(here we ignore the kv cache). Therefore, the FLOPs for a single forward pass of our method on a single task is: $FLOPs_L + 2*FLOPs_S+nBV$. Among these, only the $nBV$ term ($n \le 20$) corresponds to the additional computational cost of our method, which is much smaller compared to the previous term and can be considered negligible in the overall time. Additionally, in our efficiency analysis, as shown in Table 3, our method is only 0.008 seconds slower per sample compared to the static method, which is negligible.

### Supplementary Proof for the Fusion of Multiple SLMs Scenario (for YRcj, frHX, Ttwe)

This section mainly explains how we extend the transfer problem to multiple small models. When transferring the knowledge of multiple expert SLMs to a LLM, we consider the following two aspects: 1. The fusion of knowledge from different domain experts. 2. The transfer of knowledge from SLM to LLM, i.e., the transfer of knowledge from a single expert, which was discussed in Section 3.2. Intuitively, we first focus on the fusion of different domain experts' knowledge before performing the transfer.
 Here, we define the distribution of the combined knowledge of these small models as $J$. **Therefore, we aim to achieve $KL(P || \tilde{P})=KL(Q||J)$**

  Since solving for $J$ is difficult, we propose constraining it based on the relationship between $J$ and $\{Q_i\}$ to approximate it. Here, we can transform $KL(Q||J)$ into $KL(Q||Q_i)+C_J(Q_i)$, where $C_J(Q_i)$ is the bias function from $Q_i$ to $J$. When we approximate $J$ as the centroid of $\{Q_i\}$ on the KL-constrained plane, we can implicitly solve these bias functions. According to the definition of the centroid, $J$ can be solved by minimizing the sum of the squared distances to each point, as shown below:

  $$\arg \min_{J} \sum_{i=1}^T (KL(Q \parallel J) - KL \left(Q \parallel Q_i \right))^2$$

  Since our goal is $KL(P \parallel \tilde{P})=KL(Q||J)$, substituting this into our equation gives us our final optimization objective:


$$\arg \min_{\tilde{P}} \sum_{i=1}^T (KL(P \parallel \tilde{P}) - KL \left(Q_i \parallel Q \right))^2$$


**To prove the reasonableness of our approximation, we provide a more rigorous proof below. Our initial objective is as follows:**


$$\arg \min_{\tilde{P}} \sum_{i=1}^T (KL(\tilde{P} \parallel P) - KL(J||Q))^2$$


  By assuming $KL(Q||J)=KL(Q||Q_i)+C_J(Q_i)$, we can transform the original problem $\arg \min_{\tilde{P}}  (KL(\tilde{P} \parallel P) - KL(J||Q))^2$ into $T$ constrained optimization problems:

$$\begin{align}\arg \min_{\tilde{P}} (KL(\tilde{P} \parallel P) - KL \left(Q_i \parallel Q \right)-C_J(Q_1))^2\\\\
...\\\\
\arg \min_{\tilde{P}} (KL(\tilde{P} \parallel P) - KL \left(Q_i \parallel Q \right)-C_J(Q_T))^2\end{align}$$




After jointly optimizing them, we have:

$$\begin{align}\arg \min_{\tilde{P}} \sum_{i=1}^T (KL(\tilde{P} \parallel P) - KL \left(Q_i \parallel Q \right)-C_J(Q_i))^2\\\\
\sum_{i=1}^T (KL(\tilde{P} \parallel P) - KL \left(Q_i \parallel Q \right)-C_J(Q_i))^2 \\\\\leq \sum_{i=1}^T (KL(\tilde{P} \parallel P) - KL \left(Q_i \parallel Q \right))^2+\sum_{i=1}^TC_J(Q_i))^2\\\\
=\sum_{i=1}^T (KL(\tilde{P} \parallel P) - KL \left(Q_i \parallel Q \right))^2+C_{J-Q}\end{align}$$

Since $C_{J-Q}$ is a constant term independent of $\tilde{P}$, we can ignore it. Finally, we solve the original problem by optimizing this upper bound. When we symmetrize the terms in the KL divergence, we can obtain a similar conclusion. Therefore, in the multi-task setting, we can solve it using the following formula (As shown in Equation (6) of the paper):

  $$\arg \min_{\tilde{P}} \sum_{i=1}^T \left[(KL(P \parallel \tilde{P}) - KL \left(Q_i \parallel Q \right))^2+(KL(\tilde{P} \parallel P) - KL \left(Q \parallel  Q_i\right))^2\right]$$

---

### Decision · Program_Chairs · 2024-09-25

**Decision:**

Accept (poster)

**Comment:**

This paper proposes a dynamic logit fusion method for transferring knowledge from smaller, task-specific language models to a larger language model. The method adaptively learns the mixing weights between the models' logits at each decoding step by optimizing an objective based on KL divergence.

Reviewers found the method to be simple yet effective, achieving good results across various tasks. (YRcj: "The method is very simple... leading to better results."; frHX: "Paper well written and easy to follow... Works well in single task scenarios."). The experimental evaluation was considered comprehensive, covering multiple tasks, model sizes, and both single-task and multi-task scenarios. (YRcj: "The experiments are comprehensive... The authors also consider both single-task and multi-task scenarios."). The proposed method consistently outperformed baseline methods, including proxy tuning and full fine-tuning on smaller models, and approached the performance of directly fine-tuning the large model. (YRcj: "The results are good across all tasks... recovers a large fraction of the ceiling performance achieved by directly finetuning the large model on ground truth.")

However, the reviewers identified a few potential weakness in the initial reviews:
 - Limited theoretical justification: Initially, reviewers raised concerns about the theoretical motivation behind matching KL divergence as the objective function. (YRcj: "It is not intuitively obvious to me why matching the KL divergence is the right objective."; frHX: "Using the squared error between two KL’s is not theoretically motivated."). However, the authors addressed this concern in their rebuttal by providing a theoretical derivation based on KL-constrained reinforcement learning. (XMhg: "I thank the reviewers for their rebuttal and I appreciated their theoretical derivation. I believe that this important piece to be added to the paper.")
- Clarity of presentation: Some reviewers found certain aspects of the presentation to be unclear, particularly regarding the optimization method and the multi-task setup. (frHX: "More description of the optimization method in main text since it is a big part of the method... Missing baseline in multi-task tuning setup.") The authors addressed these concerns in their rebuttal by providing further details and clarifications.
 - Computational complexity: The method was noted to be slower at inference time compared to standard sampling. (YRcj: "As the authors mention, the proposed method is 2.5 times slower at inference time compared to standard sampling from the same model.") While the authors clarified that the increased cost is minimal compared to fine-tuning the large model, the efficiency aspect remains a potential limitation.

The paper presents a novel and effective method for weak-to-strong generalization in language models. The comprehensive experimental evaluation demonstrates the method's strong empirical performance. While some initial concerns regarding theoretical justification and computational cost were raised, the authors adequately addressed these concerns in their rebuttal.